# Hierarchical small molecule inhibition of MYST acetyltransferases

Xuemin Chen[1], Alexandra Castroverde [1], Minervo Perez [1], Ronald Holewinski[2], Kiall F. Suazo [1,2], Rashmi Karki[3], Thorkell Andresson[2], Benjamin A. Garcia [3] & Jordan L. Meier [1] ✉

MYST lysine acetyltransferases (KATs) are a class of epigenetic enzymes critical for cellular function that constitute an emerging therapeutic target in cancer. Recently, several drug-like MYST inhibitors have been reported that show promise in preclinical models as well as in clinical trials of breast cancer. Understanding the specificity of these molecules is critical for their effective use as chemical probes. Here we apply an integrated profiling strategy to systematically define the potency and selectivity of drug-like MYST KAT inhibitors. First, we use optimized chemoproteomic profiling and histone acetylation biomarkers to study the industry-developed KAT inhibitor PF-9363. This reveals dose-dependent engagement of native KAT complexes, with hierarchical inhibition following the order KAT6A/B > KAT7 » KAT8 > KAT5. This pattern of target engagement is shared by the clinical candidate PF-8144. Next, we demonstrate how PF-9363's ability to disrupt capture of MYST complex members in chemoproteomic experiments can be leveraged to identify uncharacterized candidate members of these complexes, including the transcription factor FOXK2. Applying insights from these studies to WM-8014, WM-1119 and WM-3835, which have been extensively applied in the literature as MYST probes, highlights unexpected cross-inhibition and suggests a framework for how these small molecules and biomarkers may be applied to differentiate KAT6A/B and KAT7-dependent phenotypes. Finally, we benchmark the activity of PF-9363 in the NCI-60 cell line screen, providing evidence that its ability to engage KAT8 at elevated concentrations can drive acute growth inhibition. Collectively, our studies indicate the potential for MYST KAT inhibitors, including clinical candidates, to exhibit dose-dependent target engagement reminiscent of kinase inhibitors. The assays and biomarkers described here should find broad utility in assessing selective target engagement by this inhibitor class.

Lysine acetylation is a prevalent post-translational modification (PTM) catalyzed by lysine acetyltransferase (KAT) enzymes[1]. The MYST family of KATs includes five members: KAT5, KAT6A, KAT6B, KAT7, and KAT8 (Fig. 1A)[2]. These genes are conserved from yeast to humans and are required for many critical nuclear functions, including transcription and DNA repair. The human disease relevance of these enzymes is exemplified by KAT6A and KAT6B, two closely related MYST paralogues whose translocation can form gene fusions capable of driving

[1]Chemical Biology Laboratory, National Cancer Institute, Frederick, Maryland, USA. [2]Protein Characterization Laboratory, Frederick National Laboratory for Cancer Research, Leidos Biomedical Research, Frederick, Maryland, USA. [3]Department of Biochemistry and Molecular Biophysics, Washington University School of Medicine, St. Louis, Missouri, USA. ✉e-mail: jordan.meier@nih.gov

**Fig. 1 | Inhibition of MYST acetyltransferases by drug-like small molecules.**
**A** Domain architecture of the MYST lysine acetyltransferase family. H15: H15 domain; PHD: plant homeodomain-linked zinc finger; MYST: MYST type domain with KAT activity; ED: glutamate/aspartate-rich region; SM: serine/methionine-rich domain; Chromo: chromodomain; Ser: serine-rich domain; CCHHC: Zinc finger CCHHC-type. **B** Chemical structures of drug-like MYST inhibitors of the sulfono-hydrazide (WM compounds) and benzisoxazole sulfonamide (PF-9363) class.

**C** Crystal structure of PF-9363 bound to KAT6A catalytic domain. PDB code: 8DD5. **D** Sequence alignment of ligand-binding residues in MYST enzymes. Active site residues interacting with PF-9363 are highlighted in red. Residues were shaded based on the default color scheme for ClustalX if >60% matched the following characteristics: hydrophobic (blue); positive (red); polar (green); cysteine (yellow); proline (pink); aromatic (aqua); glycine (orange).

acute myeloid leukemia (AML)[3,4]. Furthermore, KAT6A amplifications occur in 12–15% of breast cancers, and are hypothesized to facilitate tumor growth by regulating estrogen receptor (ER) signaling[5,6]. These associations have motivated the development of selective small-molecule inhibitors of MYST enzymes. Early efforts yielded WM-1119 (Fig. 1B), a drug-like KAT6A/B inhibitor capable of inducing cell cycle arrest and senescence in MYC-driven lymphoma models[7,8]. More recently, an optimized arylsulfonamide benzisoxazole PF-9363 (Fig. 1B) has been shown to disrupt ER-mediated gene expression and demonstrate in vivo efficacy in KAT6A-dependent solid tumor xenografts[9]. An analog of PF-9363 known as PF-8144 (also referred to as PF-07248144, prifetrastat) is being tested in clinical trials and has demonstrated promising pharmacokinetics, modulation of histone acetylation, and antitumor activity in ER+ breast cancer patients[10].

Nature directs MYST catalytic activity to specific histone residues using protein-protein interactions[11,12]. In contrast, selective small molecule inhibitors must differentiate the nearly identical active sites of this family (Fig. 1C, D). Structural studies indicate a highly conserved acetyl-CoA binding mode, with the cofactor engaging similar active site residues in each enzyme[13,14]. This property extends to recognition of synthetic inhibitors, as co-crystal structures reveal that MYST inhibitors engage multiple conserved active contacts, particularly backbone amide hydrogen bonds involved in binding the pyrophosphate moiety of acetyl-CoA (Fig. 1C, D)[7,9,15]. Biochemical experiments indicate MYST inhibitors target KAT6A/B over closely-related KAT7 with specificities ranging from 10- to 1000-fold (Supplementary Data 1)[7,9,15]. This variable selectivity window is reminiscent of kinase inhibitors, which similarly must differentiate between closely related active sites and often display dose-dependent selectivity[16]. Direct biochemical and cellular comparisons of sulfonohydrazide and benzisoxazole sulfonamide MYST KAT inhibitors have not been reported. An additional challenge in comparing literature selectivity profiles of MYST inhibitors is that the reported half-maximal inhibitor concentration (IC$_{50}$) measurements arise from biochemical assays whose conditions often differ. In particular, the use of full-length enzymes or reconstituted MYST complexes—in contrast with excised catalytic domains—is uncommon[12]. To help researchers accurately deploy MYST inhibitors as chemical probes and clinical agents, there remains an unmet need to: (1) define their proteome-wide target engagement profiles, (2)

consolidate our understanding of what histone acetylations can be used as biomarkers of target engagement in cells and tissues, (3) analyze the comparative specificity and potency of MYST inhibitors across as broad a spectrum of chemotypes as possible, and (4) characterize the phenotypic relevance and any polypharmacology they may display.

Here, we present a comparative analysis of drug-like MYST acetyltransferase inhibitors. First, we use chemoproteomics to validate the ability of the industry-developed KAT6A/B inhibitor PF-9363 to engage MYST acetyltransferases within their native multiprotein complexes. This provides evidence that, in addition to KAT6A/B, elevated concentrations of PF-9363 can hierarchically occupy KAT7, KAT8, and KAT5 in the context of their native multiprotein complexes, a property shared by the clinical candidate PF-8144. The ability of PF-9363 to antagonize chemoproteomic capture of MYST complexes, in combination with clustering and structure prediction, is used to identify the transcription factor FOXK2 as a candidate MYST complex interactor. Our knowledge of PF-9363's dose-dependent target engagement allows us to validate specific histone acetylation biomarkers for KAT6, KAT7, KAT8, and KAT5, which we apply to study the selectivity of the sulfonohydrazide MYST inhibitors WM-8014, WM-1119 and WM-3835, as well as the mechanisms of acute growth inhibition caused by PF-9363 in the NCI-60 cell line screen. Our studies indicate the highly similar active sites of MYST KAT enzymes render them susceptible to dose-dependent target engagement, similar to how ligands engage the ATP-binding sites of closely related kinases, and highlight general methods to evaluate selective versus multi-enzyme inhibition in proteomic and cellular settings.

## Results
### Chemoproteomics reveals dose-dependent MYST complex engagement by PF-9363
Structural studies indicate MYST inhibitors achieve active site binding by mimicking the pyrophosphate moiety of acetyl-CoA. However, while the activities of these molecules against purified MYST enzymes in biochemical assays are well-established, their ability to selectively engage native MYST complexes in a proteomic mileau remains uncharacterized. To address this, we applied a chemoproteomic strategy to monitor MYST inhibitor selectivity (Fig. 2A)[17,18]. This

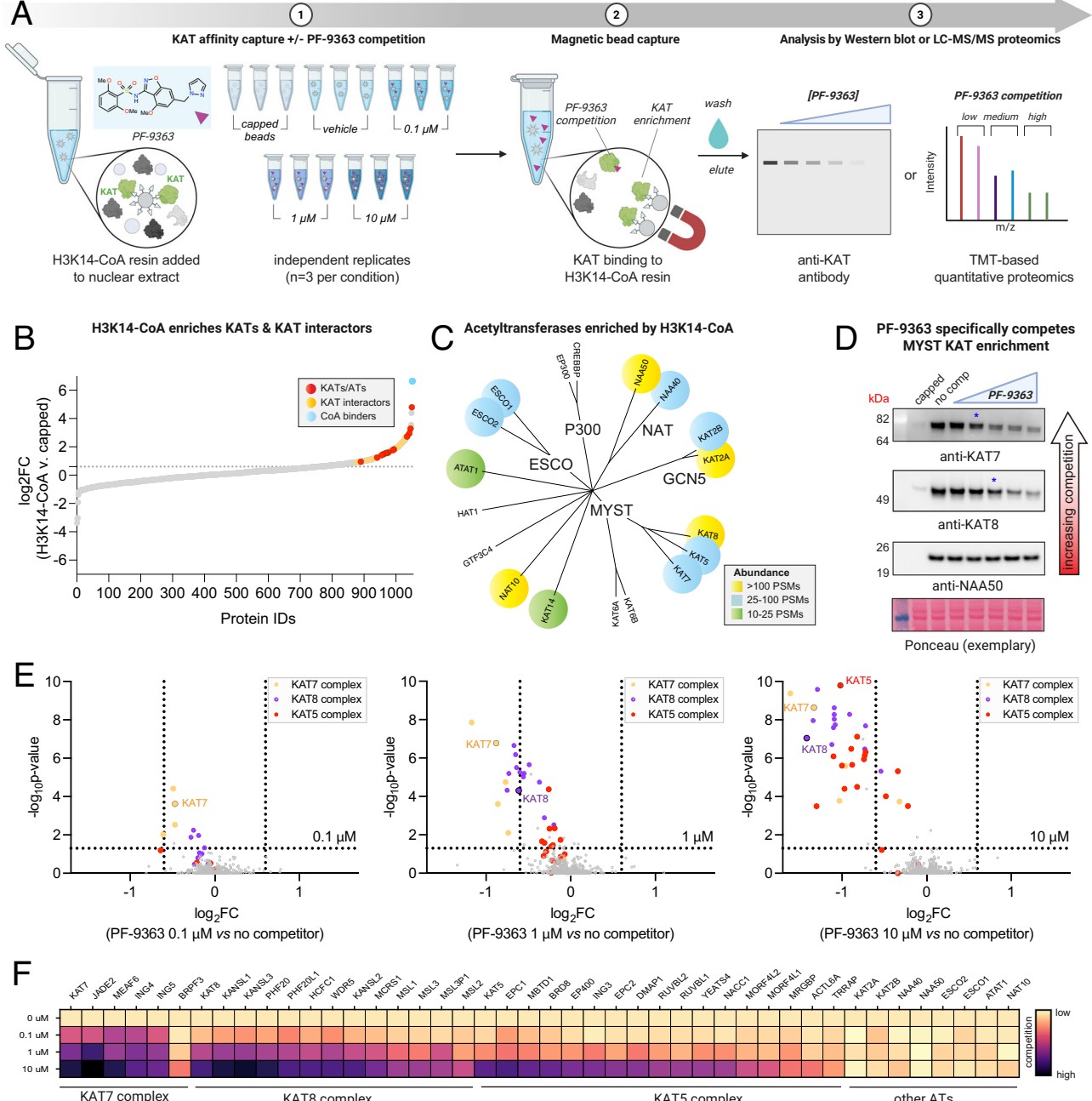

**Fig. 2 | Competitive chemoproteomics reveals dose-dependent MYST occupancy by PF-9363. A** Schematic of an optimized competitive chemoproteomic profiling strategy for KAT enzymes. Created in BioRender. Meier, J. (2026) https://BioRender.com/sg3kn8x. **B** LC-MS/MS analysis of enriched proteins. Proteins above the dashed line represent targets specifically enriched by H3K14-CoA resin (log2FC (H3K14-CoA vs capped) > 0.6, *p*-value < 0.05). Red: KATs/ATs; Yellow: KAT interactors; Blue: Other CoA binders; Gray: all other proteins. **C** Phylogenetic tree depicting KATs enriched by H3K14-CoA resin by proteomics. Proteins are color-coded based on peptide-spectrum match (PSM) abundance. **D** PF-9363 dose-dependently competes enrichment of MYST KATs. HeLa nuclear extracts were pre-incubated with escalating concentrations of PF-9363 competitor: 0.01, 0.1, 1, 10, 30 μM (2 h, 4 °C). Ethanolamine-capped beads were used to assess non-specific binding. 3 biological replicates were performed for each condition. **E** Proteome-wide competition analysis of PF-9363 competition in HeLa nuclear extracts (*n* = 3 biological replicates). Nuclear extracts were pre-incubated at the specified concentration (2 h, 4 °C) prior to KAT affinity capture. MYST KATs and interactors are color-coded according to the complex. **F** Heatmap summary of chemoproteomic competitions of PF-9363 with ATs and associated complex members in HeLa nuclear extracts (*n* = 3 biological replicates). Light colors indicate low engagement, dark colors indicate greater occupancy.

method uses a resin-immobilized CoA analog to afford sensitive capture of KATs and other CoA-binding proteins via their active sites. Pre-incubation of proteomes with active-site competitive small molecules blocks KAT capture in a dose-dependent manner, providing a readout of inhibitor potency and selectivity[19,20]. To facilitate MYST inhibitor profiling, we optimized our previously reported protocols, leveraging

nuclear extracts, capture by an H3K14-CoA bisubstrate, and magnetic bead separation to improve throughput and sensitivity (Fig. S1). This enabled enrichment of MYST KATs (KAT5, KAT7, KAT8), associated complex members, and many non-MYST (KAT2A, KAT2B, NAA50, NAA40, NAT10, ESCO1, ESCO2, ATAT1) acetyltransferases (Fig. 2B, C). Unfortunately, this method did not sample KAT6A or KAT6B, the most

potent targets of PF-9363 in biochemical assays[9]. Whole proteome profiling of HeLa nuclear extracts did not identify KAT6A/B, suggesting these proteins are present at low copy numbers or poorly extracted from chromatin using traditional fractionation protocols (Fig. S2A, Supplementary Data 2). Comparing expression of MYST genes in DepMap indicated KAT6A/B are generally more weakly expressed than other family members (Fig. S2B). To address this 'blind spot' and assess the ability of chemoproteomics to assess selectivity across the MYST family, we performed a pilot in which we spiked recombinant KAT6A into nuclear extracts and measured competition by PF-9363 or PF-8144 at a single concentration (1 μM). KAT6A was the most potently competed CoA-binding protein, followed by KAT7 (Supplementary Data 3; Fig. S3). Having confirmed the literature, we next set out to determine the dose-dependent effects of PF-9363 across a broader concentration gradient. Using endogenous nuclear extracts, we confirmed that pre-incubation of proteomes with PF-9363 reduced the capture of KAT7, as assessed by western blot (Fig. 2D). Proteomic analysis of capture experiments confirmed KAT7 was the most significantly competed CoA-binding protein at low concentrations of PF-9363 (0.1 μM; Fig. 2E, Supplementary Data 4). This is consistent with prior findings indicating that after KAT6A/B, KAT7 is PF-9363's second-most sensitive MYST target[9].

KAT7 operates within a quaternary complex containing MEAF6, an ING protein (ING4/5), and either JADE (1/2/3) or BRPF (1/2/3) subunits[21]. At low (0.1 μM) concentration, PF-9363 also caused co-competition of JADE2, MEAF6, and ING4/5, indicative of engagement of the JADE complex (Fig. 2E, F, Supplementary Data 4). BRPF3 was also enriched by H3K14-CoA resin, potentially due to capture of a BRPF complex. In contrast to the JADE complex, BRPF3 displayed competition only at high concentrations of PF-9363 (10 μM, Fig. 2F). We confirmed the differential competition of JADE2 and BRPF3 capture using WM-3835, a putative KAT7-selective inhibitor (Supplementary Data 5). Our chemoproteomic assay cannot distinguish whether this reflects preferential capture of the BRPF complex by the affinity resin, preferential engagement of the JADE complex by PF-9363, or association of BRPF3 with another MYST complex.

Exploring competition profiles observed at higher PF-9363 concentrations (1 and 10 μM) we find PF-9363 competed capture of all CORUM-annotated members of the KAT8-containing MSL (KAT8, MSL1/2/3, MSL3P1) and NSL complexes (KAT8, KANSL1/2/3, HCFC1, MCRS1, OGT, PHF20, PHF20L1, WDR5)[22] as well as most members of the KAT5-containing NuA4 complex (MEAF6, EPC1/2, DMAP1, ING3, MBTD1, YEATS4, VPS72, EP400, BRD8, RUVBL1/2, MORF4L1/2, ACTL6A, MRGBP, TRRAP; Fig. 2F)[23]. The sensitivity of KAT complexes to PF-9363 follows the general order KAT7 > KAT8 > KAT5. Proteins belonging to multiple KAT complexes are characterized by distinct competition profiles. For example, MEAF6 (KAT5/KAT7) stands out amongst KAT5 complex members in that it also displays potent competition at low concentrations (0.1 μM), presumably reflecting competition of its KAT7 JADE complex-associated fraction. In contrast, TRRAP displays less competition than most NuA4 complex members (e.g., KAT5 itself), reflecting the inability of PF-9363 to compete with capture of this protein residing in the KAT2-associated STAGA complex (Fig. 2F)[24,25]. These studies establish the ability of the benzisoxazole sulfonamide ligand PF-9363 to selectively occupy the catalytic CoA-binding domain of MYST acetyltransferase complexes in complex proteomes.

### Applying chemoproteomics to identify previously unknown candidate MYST interactors

Applying MYST inhibitors as precision chemical probes requires not only defining their selectivity, but also the composition of the protein assemblies they engage. Encouraged by the ability of chemoproteomics to identify annotated members of MYST complexes, we next sought to assess whether any unannotated proteins displayed similar patterns of PF-9363-mediated competition, potentially indicative of uncharacterized interactors. To explore this in a systematic manner, we analyzed our chemoproteomic data using an approach similar to our previously reported CATNIP method (Fig. 3A)[17]. Briefly, fold-change values generated from capture experiments conducted in the presence of 0, 0.1, 1, and 10 μM PF-9363 were transformed[26], plotted in two dimensions, and subjected to k-means clustering[27]. Five groups of proteins were identified, with two forming a relatively tight cluster and three more dispersed (Supplementary Data 6). Inspection of individual proteins within each cluster revealed characteristic dose-response signatures (Fig. 3B). The capture of proteins in clusters 2 and 3 were antagonized by PF-9363 in a dose-dependent fashion, with cluster 2 being slightly more sensitive than 3 (Fig. 3B, **top/middle**). Proteins in other clusters, including acetyltransferases such as NAT10, ESCO1, and KAT2B, were relatively insensitive (Fig. 3B, **bottom**). Clusters 2 and 3 encompass 35 proteins in total, 33 of which are known members of KAT7 (BRPF/JADE), KAT8 (NSL/MSL), or KAT5 (NuA4) complexes (Fig. 3C). KAT7 JADE complex members were found exclusively in cluster 2, while other MYST complex members were distributed across both. The variant histone H2.AZ (Uniprot: H2AV) was an unexpected cluster member that displays dose-dependent competition by PF-9363 (Supplementary Data 4). Previous studies have found that in addition to histone acetylation, KAT5-containing complexes also mediate ATP-dependent exchange of H2A-H2B for H2A.Z-H2B[28], providing a rationale for association with NuA4. The other unanticipated protein found to show a similar pattern of competition to MYST complex members in chemoproteomic experiments was the transcription factor FOXK2 (Fig. 3C).

FOXK2 is a member of the forkhead box transcription factor family, known for its evolutionary conserved winged helix DNA binding domains and wide-ranging involvement in biological processes[29,30]. Circumstantial evidence suggests FOXK2 requires partner proteins to carry out its functions. FOXK2 binds the interleukin-2 (*IL2*) promoter but may enable *IL2* expression by altering the structure of chromatin[31]. Similarly, genome-wide binding studies indicate association of FOXK2 with forkhead consensus sequences is necessary but not sufficient for target gene expression[32]. To better understand the ability of FOXK2 to interact with MYST complexes, we performed an in silico interaction screen using AlphaPulldown[33,34]. AlphaPulldown is an iterative implementation of AlphaFold2 that uses a ColabFold Search to predict interactions between a single bait protein (e.g. FOXK2) and a large set of potential binding partners, allowing high-throughput discovery of candidate protein interactors (Fig. 3D). We applied this to predict binding conformations of FOXK2 in complex with all 34 proteins showing similar chemoproteomic competition profiles (clusters 2 and 3, above) and then scored them based on multiple parameters including ipTM (Interface Predicted TM-Score), normalized LIS (Local Interaction Score) and LIA (Local Interaction Area)[35], and mpDockQ/pDockQ (Supplementary Data 7)[36]. Additionally, we also introduced a composite score that sums the normalized min-max LIA, LIS, and mpDockQ/pDockQ weighted by a filter-based confidence coefficient (Supplementary Information). Ranking these interactors by composite score revealed the top hits to be OGT and WDR5 (Fig. 3D). In both cases, interactions were predicted to occur through a region of FOXK2 that was largely disordered (Fig. S4). Given the intrinsic uncertainty in predicting interactions of disordered regions, we experimentally evaluated the ability of OGT and WDR5 to interact with FOXK2 in co-immunoprecipitation assays (Fig. 3E, F). Ectopically-expressed FOXK2-FLAG captured using an anti-FLAG resin was able to co-enrich a Myc-tagged OGT using both basal and high salt wash conditions (Fig. 3E). An identical procedure was not able to co-precipitate WDR5-Myc using high salt wash conditions (Fig. 3F). STRING analysis revealed OGT has been previously linked to FOXK2 as a member of the polycomb repressive deubiquitinase (PR-DUB) complex[37,38]. Interestingly, FLAG-FOXK2 by itself was able to co-immunoprecipitate endogenous KAT8.

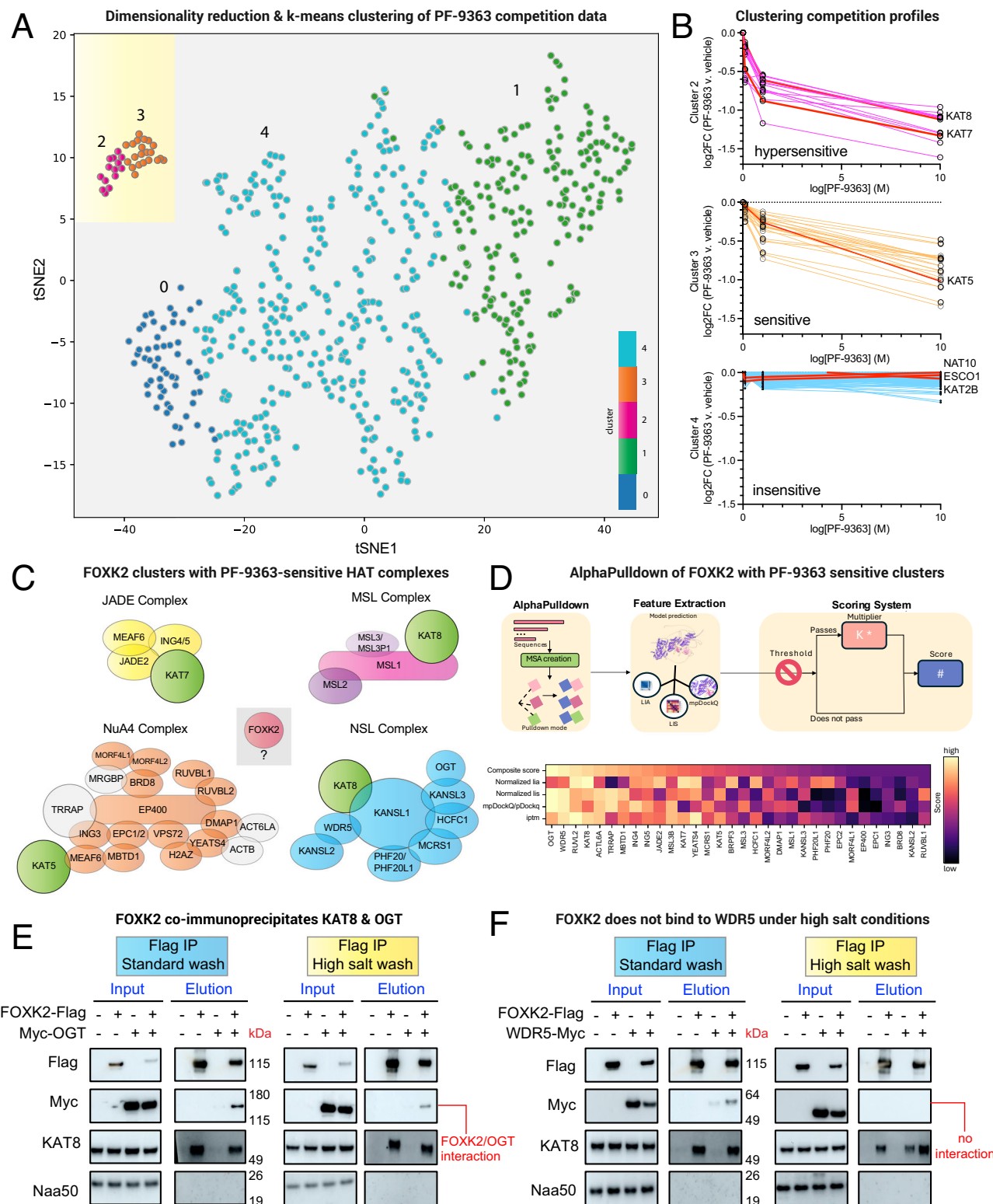

**Fig. 3 | Applying chemoproteomic competition to identify previously unknown candidate MYST interactors. A** t-SNE clustering of proteins based on PF-9363 chemoproteomic competition profile (PF-9363 = 0, 0.1, 1, 10 μM). **B** Dose-response profiles of PF-9363 competition clusters. Colored lines indicate the capture profiles of individual proteins at each concentration of PF-9363 competitor. Red lines represent the capture profiles of labeled KAT or AT enzymes. **C** PF-9363 sensitive proteins identified by this analysis. Green: KAT proteins; Orange: NuA4 complex, Blue: NSL complex; Yellow: JADE complex; Gray: MYST complex proteins not clustered by t-SNE; Red: candidate MYST interactors that were previously unknown. **D** Top: Schematic of interaction screening of FOXK2/PF-9363-

sensitive proteins by AlphaPulldown. Created in BioRender. Meier, J. (2026) https://BioRender.com/r25wv0c. Bottom: Ranking of predicted FOXK2-interacting proteins by AlphaFold metrics and composite score. Light colors indicate stronger prediction confidence, dark colors indicate weaker prediction confidence. **E** Co-immunoprecipitation of FLAG-FOXK2 and Myc-OGT (*n* = 2 biological replicates). Left: standard wash; Right: High salt wash. OGT interaction persists in the presence of salt. **F** Co-immunoprecipitation of FLAG-FOXK2 and WDR5-Myc (*n* = 2 biological replicates). Left: standard wash; Right: High salt wash. WDR5 interactions are salt-labile, suggestive of indirect or low-affinity contacts. In both cases, FLAG-FOXK2 is found to co-immunoprecipitate endogenous KAT8.

Previous studies have observed that members of the NSL complex (KANSL1/KANSL1L) co-immunoprecipitate with FOXK2[39], further supporting the ability of this transcription factor to fractionally occupy the MYST complex in pulldown experiments. Future studies will be required to understand the relevance of the FOXK2 interaction to NSL-dependent biology. These studies demonstrate a strategy for leveraging chemoproteomics and MYST selective inhibitors to identify candidate interaction partners of multiprotein KAT complexes.

### PF-9363 elicits dose-dependent changes in MYST-regulated histone acetylation

Our chemoproteomic data indicate that high concentrations of PF-9363 are capable of biochemically occupying the active site of several MYST complexes in nuclear extracts. This raises the question: to what extent is this phenomenon biologically relevant (e.g., do high concentrations of PF-9363 hierarchically engage KAT6, KAT7, KAT8, and KAT5 in cells)? To understand how PF-9363 impacts global histone modifications in a relatively unbiased manner, we profiled its effects across an escalating gradient of concentrations using a bottom-up proteomic method[40]. In this approach, MCF-7 breast cancer cells are treated with increasing doses of PF-9363 (0.1, 1, 10, 30 μM) or vehicle (DMSO) for 24 h (Fig. 4A). Histones are then extracted from cell pellets and chemically derivatized with propionic anhydride, digested with trypsin, subjected to a second round of propionylation, and analyzed by LC-MS/MS. Measuring the relative intensity of modified versus unmodified forms of individual histone peptides enables simultaneous monitoring of multiple potential biomarkers of MYST inhibition. To minimize false positives, we focused on abundant and reproducible modifications (average stoichiometry of greater than 1% across triplicate samples, standard deviation less than or equal to the average measurement). To account for potential histone modification crosstalk, changes in both acetylated and methylated peptides were evaluated. 38 modified peptides passed this reproducible detection threshold including several established KAT biomarkers[41] including H3K9Ac (KAT2A/B), H3K18Ac (KAT3A/B), H3K14Ac (KAT7), H3K23Ac (KAT6A/B), H4K16Ac (KAT8), and H2A.ZK4/K7Ac (KAT5; Fig. 4B). Eight peptides displayed a ≥ 2-fold change in abundance between low-dose (0.1 μM) and high-dose (30 μM) PF-9363 treatment conditions, all of which contained acetylated lysine residues (Fig. 4D, Supplementary Data 8). No exclusively methylated peptides were significantly changed by PF-9363 treatment, a phenomenon we confirmed via western blot analysis of H3K27 and H3K79 sites (Fig. S5). This is in line with prior genomic analyses of PF-9363, which similarly did not observe changes in the genome-wide distribution of H3K4 or H3K27 methylation[9]. These studies indicate that the acute effects of PF-9363 on abundant histone marks in MCF-7 cells are limited to inhibition of acetylation.

Modified peptides whose abundance was reduced by PF-9363 treatment correspond to six acetyllysine positions: H3K23Ac, H3K14Ac, H4K16Ac, H2A.Z4, H2A.Z15, and H2A.Z11 (Supplementary Data 8, Fig. 4B). Dose-response profiling revealed that across these acetylations, H3K23Ac was most sensitive to PF-9363, displaying an ~80% reduction upon 0.1 μM treatment (Fig. 4C, D). Genetic perturbations have unambiguously determined the dependence of H3K23Ac on KAT6A/B[9,41]. The H3K23Ac biomarker critically complements chemoproteomics by allowing sensitive assessment of KAT6A and KAT6B activity, whose active site occupancy is otherwise difficult to assess. Dose-response profiling confirmed that higher concentrations of PF-9363 inhibit H3K14Ac (IC$_{50}$ ~ 1 μM) followed by H4K16Ac (IC$_{50}$ ~ 10 μM), consistent with sequential engagement of KAT7 and KAT8, respectively (Fig. 4C, D). PF-9363 was not found to modulate H3K9Ac or H3K18Ac at any concentration, indicating these marks are not relevant biomarkers of MYST activity. This is significant since several studies have used H3K9Ac as a proxy for KAT6A/B inhibition (Supplementary

Data 8)[8,42], even though its regulation is most commonly attributed to KAT2A/B[41]. Since LC-MS/MS assays require specialized equipment and technical expertise, we also validated a panel of histone acetylation antibodies that recapitulate the dose-dependent inhibition of MYST targets by western blot (Fig. 4E). These studies confirm the dose-dependent pharmacology of PF-9363 and establish a panel of biomarkers for monitoring MYST engagement in living cells.

### Assessing target engagement by orthogonal MYST inhibitor chemotypes

Having established an assay to evaluate the cellular selectivity of MYST inhibitors, we extended it to additional compounds (Fig. 5A). PF-8144 is an analog of PF-9363, currently being assessed in Phase 1 clinical trials, whose structure was disclosed in November 2025[10]. The sulfonohydrazide-based compounds WM-8014, WM-1119, and WM-3835 are commonly used chemical probes of MYST enzymes. The first member of this compound class to be disclosed was WM-8014, a putative KAT6A/B inhibitor that induces cell cycle arrest and senescence in lymphoma models[8]. A more bioavailable analog, WM-1119, was subsequently developed, and reported to have increased potency and selectivity for KAT6A/B[7,8]. A third sulfonohydrazide, WM-3835, has been used as a chemical probe of KAT7 and phenocopies aspects of its deletion in AML models[14]. MOZ-IN-3 mimics the connectivity of the sulfonohydrazide using a methyl sulfonamide scaffold and has the unusual property of inhibiting KAT6A with 277-fold selectivity over KAT6B in biochemical assays[43]. A direct comparison of the dose-dependent effects of these inhibitors across MYST-dependent biomarkers has never been performed.

To enable this comparison, we treated MCF-7 cells with each compound across a concentration range spanning 0.1 to 30 μM, including PF-9363 as a reference. Engagement of specific MYST enzymes was assessed using the panel of histone acetylation biomarkers defined above. The clinical lead PF-8144 displayed a nearly identical inhibition profile to PF-9363, with slightly decreased inhibition of KAT8 (H4K16Ac) at elevated concentrations (≥10 μM). Shifting our attention to the first-generation sulfonohydrazide WM-8014, we observed it was less potent than PF-9363 but afforded similarly complete inhibition of KAT6A/B—as assessed by inhibition of H3K23Ac—at 1 μM (Fig. 5B). At higher doses (10–30 μM), WM-8014 also inhibited KAT7 (H3K14Ac), consistent with prior reports. WM-1119 inhibited H3K23Ac with potency similar to PF-9363, but surprisingly showed little inhibition of KAT7 even at the highest concentrations tested (30 μM; Fig. 5B). This observation highlights WM-1119 as a highly selective cellular probe of KAT6A/B activity. While WM-3835 has been applied to study KAT7, its initial publication does allude to its ability to inhibit both KAT6A/B and KAT7[14,44]. Consistent with this, cellular profiling indicates ~100-fold more potent inhibition of KAT6A/B (H3K23Ac) than KAT7 (H3K14Ac) in the MCF-7 model (Fig. 5B). All commercial vendors currently market WM-3835 as a KAT7 inhibitor. Our results highlight a need for caution when interpreting mechanisms arising from WM-3835 treatment and suggest KAT7-dependent phenotypes may be better defined by contrasting the effects of KAT6-selective inhibitors (WM-1119) with dual KAT6/7 antagonists (WM-8014/WM-3835). Finally, MOZ-IN-3 showed no effects on MYST-dependent biomarkers in MCF-7 cells, possibly indicative of limited uptake. Overall, these studies demonstrate how dose-dependent biomarker analysis can inform the application of chemical probes of MYST acetyltransferase enzymes.

### Engagement of KAT8 by high-dose PF-9363 correlates with acute growth inhibition

One implication of the data above is that the ability of MYST inhibitors to hierarchically engage multiple targets can act as a relevant driver of their physiological effects. Specifically, even though KAT6A/B is the primary target of WM-3835, its ability to secondarily engage KAT7

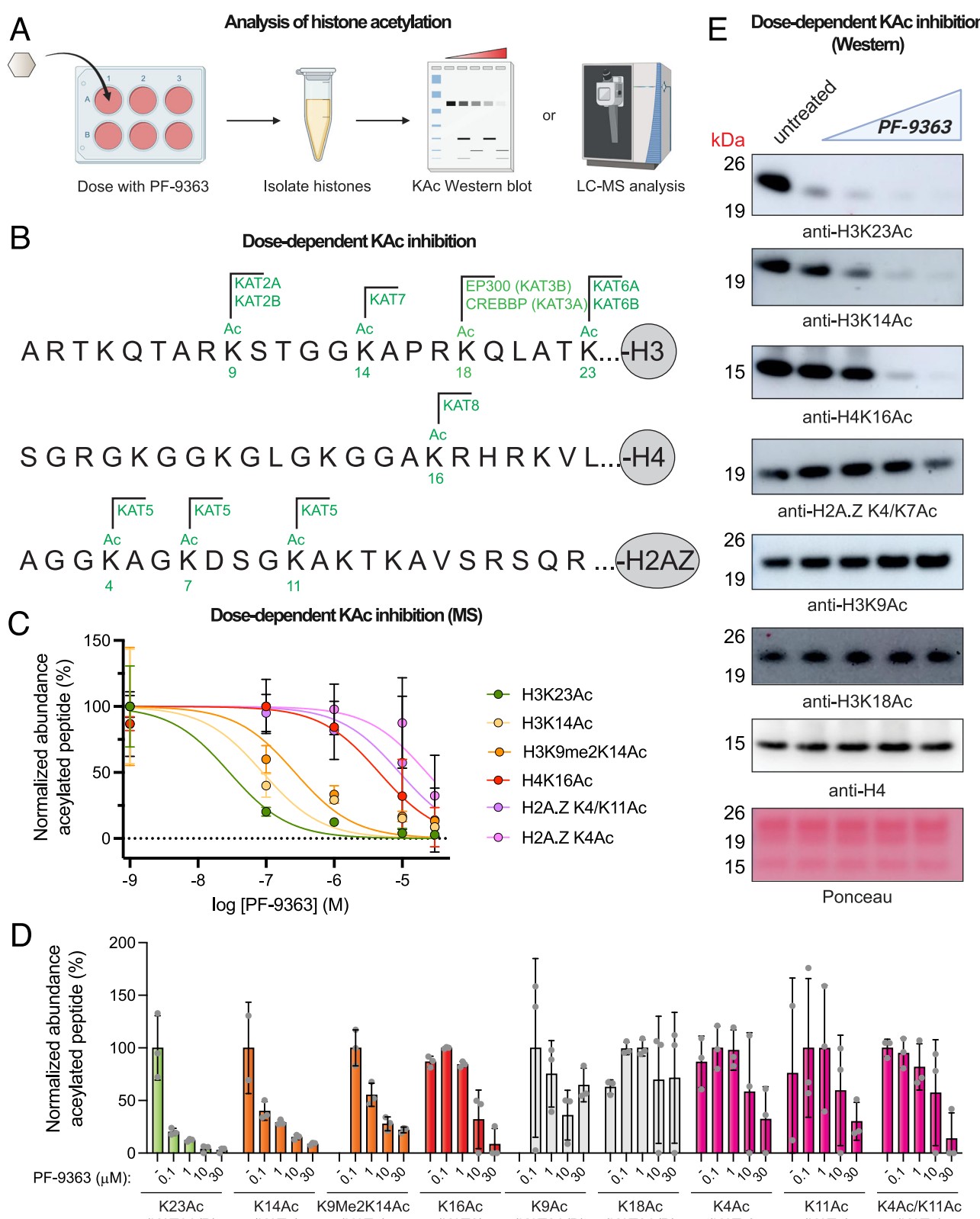

**Fig. 4 | Histone biomarker analysis reveals hierarchical inhibition of MYST acetyltransferases by PF-9363. A** Schematic of the histone biomarker assay used to assess PF-9363 inhibition of KAT-mediated acetylation in MCF-7 cells. Created in BioRender. Meier, J. (2026) https://BioRender.com/8g0oa0i. **B** Annotated histone acetylation biomarkers of MYST and non-MYST KAT enzymes. CREBBP and EP300 are used interchangeably with their synonyms KAT3A and KAT3B. **C** Dose-dependent inhibition of histone acetylation biomarkers sensitive to PF-9363 as assessed by bottom-up LC-MS/MS proteomics (fold-change ≥ 2, 0 v. 30 μM PF-9363, $n = 3$ biological replicates). Data are presented as mean values ± SD. **D** LC-MS/MS analysis of histone acetylation biomarkers upon treatment with escalating doses of PF-9363 ($n = 3$ biological replicates). Data are presented as mean values ± SD. PF-9363-sensitive markers are colored according to the MYST complex; insensitive markers are colored gray. **E** Facilte monitoring of KAT-regulated histone acetylation in response to PF-9363 by western blot (MCF-7 cells, 24 h treatment, 0, 0.1, 1, 10, 30 μM). Data is representative of $n = 2$ biological replicates.

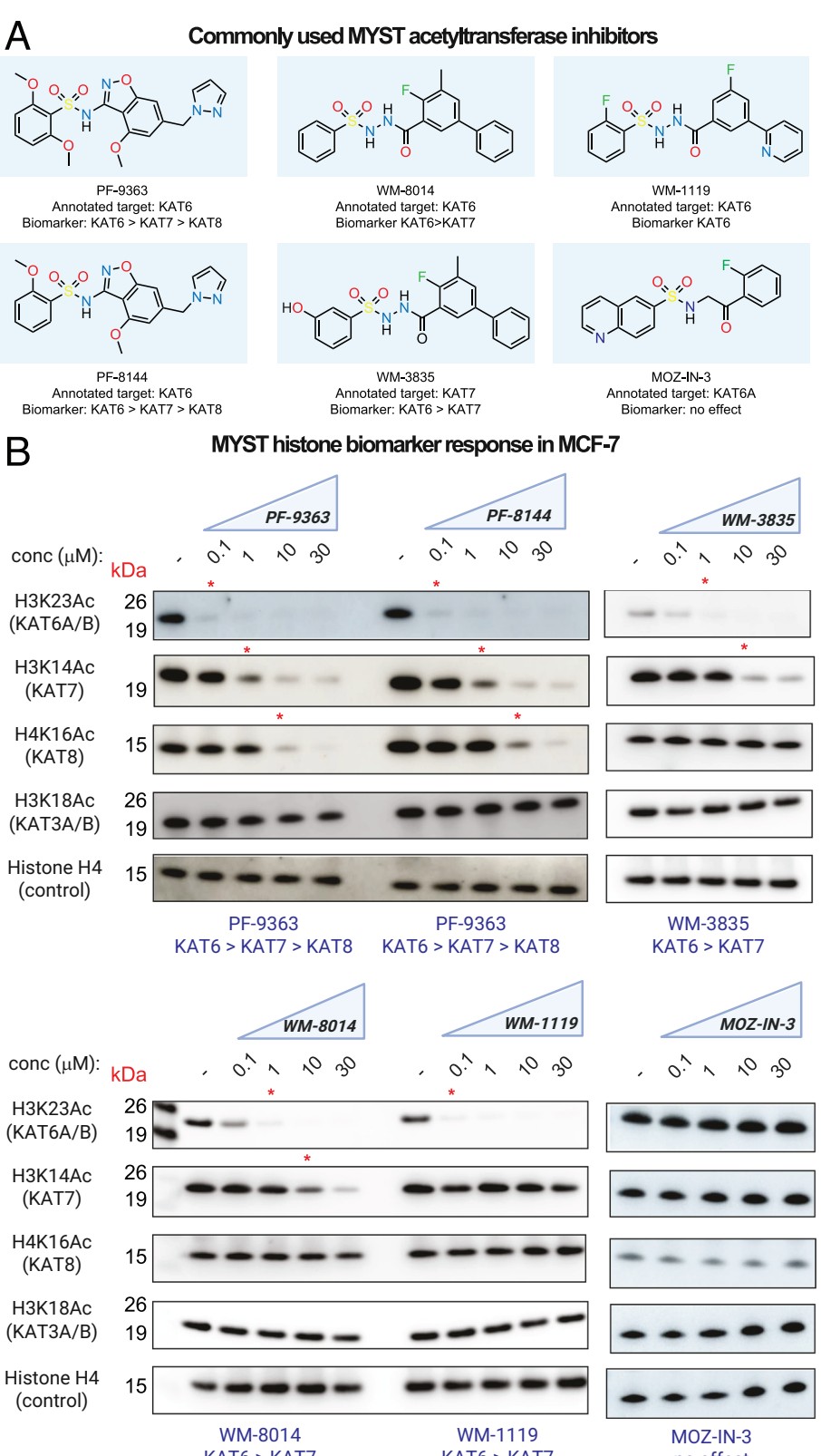

**Fig. 5 | Comparative cellular analysis of orthogonal MYST inhibitor chemotypes. A** Chemical structures of commonly used MYST KAT inhibitors. Annotated target: KAT enzyme specified as the target of each inhibitor in scientific or commercial vendor literature. Biomarker: KAT-regulated histone acetylation biomarker found to be modulated by each compound in this study. **B** Cellular response of histone acetylation biomarkers for KAT6, KAT7, KAT8, and KAT5 to treatment with drug-like MYST acetyltransferase inhibitors. MCF-7 cells were treated with escalating dosages (0.1, 1, 10, 30 μM.) of each compound for 24 h. Red asterisks indicate PF-9363 concentrations where obvious inhibition of a given mark is observed. Data is representative of $n = 2$ biological replicates.

allows it to phenocopy aspects of gene knockout[14]. Put another way, while WM-3835 is not a specific chemical probe of KAT7, it can usefully modulate KAT7-dependent phenotypes. Considering the biomarker profile of PF-9363 (Fig. 5B), we wondered if this strategy could be similarly harnessed to alter a KAT8-dependent phenotype. As a proof-of-concept, we focused on acute growth inhibition. Our rationale was that, when used to inhibit KAT6A/B and/or KAT7, MYST inhibitors require prolonged administration (~8–21 days) to suppress senescence[8] or growth[9,45]. Unlike KAT6A/B and KAT7, KAT8 is a universally essential enzyme. Thus, upon treating cells with concentrations of PF-9363 sufficient to engage KAT8, we would expect acute growth defects that are relatively independent of cell lineage compared to other KAT inhibitors.

To explore the hypothesis that KAT8 engagement can drive acute growth inhibition phenotypes, we evaluated the sensitivity of the NCI-60 cell line panel to PF-9363 across five doses ranging from 0.1 to 100 μM over 48 h. Concentrations of PF-9363 expected to primarily effect KAT6 and KAT7 (≤1 μM) were generally well-tolerated by the NCI-60 cell panel, with half-maximal growth inhibition ($GI_{50}$) for most cell lines lying in the 10–100 μM range, where KAT8 occupancy may also occur (Figs. 6A, S6). To assess whether pan-MYST engagement highlights any notable dependencies relative to other KAT inhibitors, we compared PF-9363 growth inhibition with that caused by CPI-1612[46,47], a potent inhibitor of EP300 and CREBBP (KAT3A/B). In most cases, CPI-1612 was a much more potent suppressor of cell growth (Fig. 6A). One exception came in BT-549, a triple negative breast cancer model where PF-9363 showed greater inhibition at high concentrations (10 and 100 μM; Fig. 6B). Given our notion that acute proliferation defects may reflect KAT8

inhibition, we sought to evaluate the effects of PF-9363 in this model in greater detail.

Analysis of DepMap-derived CRISPR screening data generated in breast cancer models indicates that BT-549 proliferation is largely independent of EP300 and CREBBP, modestly inhibited by knockout of KAT6 or KAT7, and highly dependent on KAT8, consistent with the latter's designation as an essential gene (Fig. 6C)[48,49]. To validate the NCI-60 results we carried out dose-response cytoxocity assays and determined PF-9363 inhibits BT-549 with a $GI_{50}$ of ~7 μM (Fig. 6D). Analysis of histone biomarkers indicated that near complete inhibiton of KAT6A/B (H3K23Ac), KAT7 (H3K14Ac) and KAT8 (H4K16Ac) was achieved at 10 μM which lies close to the $GI_{50}$ dose (Fig. 6E). In contrast, KAT3 (H3K18Ac) and KAT5 (H2A.ZK4/K7Ac) were unaffected. These effects are consistent with inhibition of BT-549 cell growth occurring through a KAT8-dependent mechanism, although unambiguous confirmation will require the development of mutants of KAT8 resistant to PF-9363, an aim beyond the scope of this study. Helin and coworkers[50] have suggested a therapeutic window may exist for targeting KAT8 in KAT8-low tumors, as in these settings KAT8 preferentially associates with the essential NSL complex, whereas in healthy tissues the MSL complex provides a reservoir of KAT8 that can be inhibited without impacting fitness. Evaluation of this strategy may benefit from experiments in which the effects of high concentrations of WM-1119 (KAT6), WM-3835 (KAT6/KAT7), and PF-9363 (KAT6/KAT7/KAT8) are contrasted, allowing KAT8's contribution to be defined. These studies provide a strategy for pharmacological modulation of KAT8 catalytic activity and demonstrate how a knowledge of MYST inhibitor pharmacology can guide experimental interpretation and design.

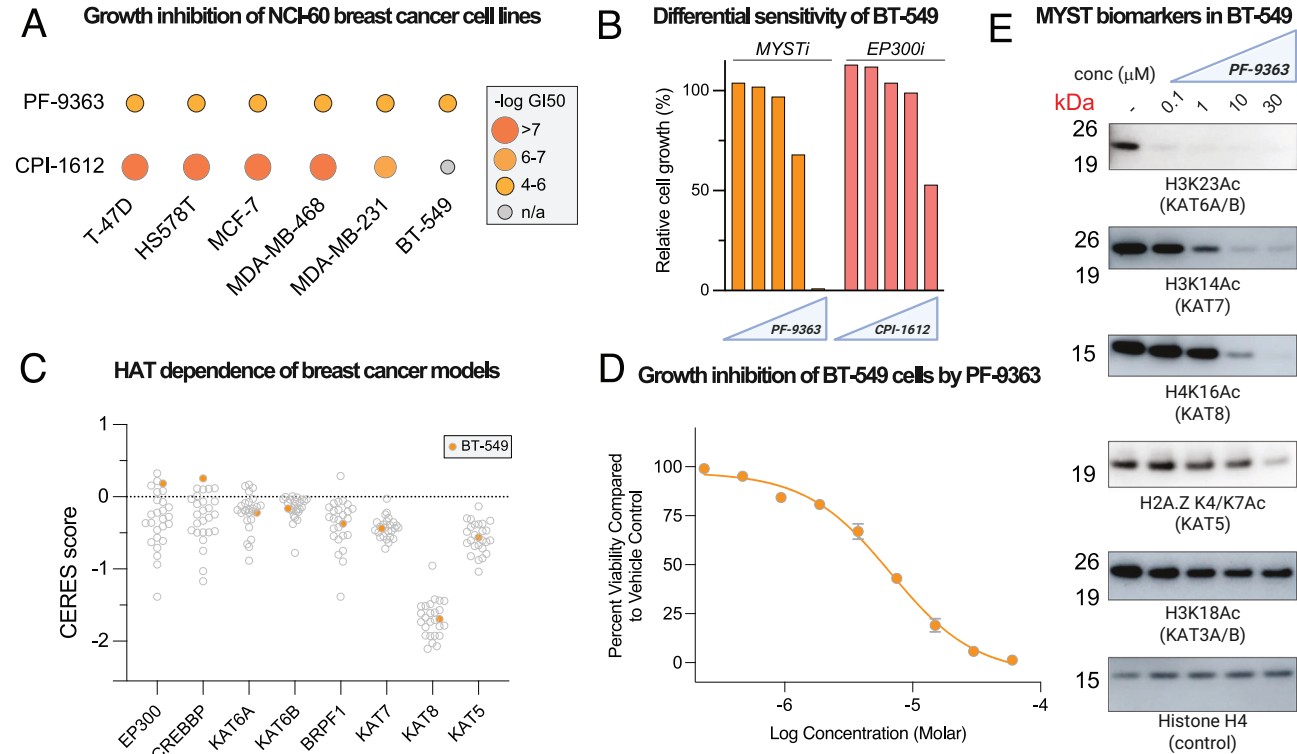

**Fig. 6 | Pan-MYST engagement correlates with acute growth inhibition. A** Half-maximal growth inhibition (GI50) values for PF-9363 and CPI-1612 across breast cancer models in the NCI-60 cell line panel (48 h). **B** Relative cell growth of BT-549 cells in the presence of escalating doses of PF-9363 or CPI-1612 (0.01, 0.1, 1, 10, 100 μM, 48 h) as assessed by sulforhodamine B assay in the NCI-60 cell line panel. Pilot screen, *n* = 1. **C** Dependence of breast cancer cell lines on non-MYST KATs (EP300, CREBBP), MYST KATs (KAT6A, KAT6B, KAT7, KAT8, KAT5), and a MYST KAT complex member (BRPF1) derived from on DepMap CRISPR screening data. BT-549 is highlighted in orange. A lower CERES score indicates a greater dependence of the cell line on the specified gene. **D** Dose-dependent growth inhibition of BT-549 cells by PF-9363 (ATP-Glo assay, 72 h, *n* = 4 replicates). Data are presented as mean values +/- SD. **E** Dose-dependent inhibition of MYST KAT biomarkers by PF-9363 in BT-549 cells (24 h, 0, 0.1, 1, 10, 30 μM). Data is representative of *n* = 2 biological replicates.

## Discussion

Targeting lysine acetylation via inhibition of KAT activity is an emerging paradigm in oncology[44], with several compounds now in clinical evaluation. Critical to identifying therapeutic contexts for these agents is the proper interpretation of their preclinical effects. Towards that end, here we report a comparative analysis of drug-like MYST acetyltransferase inhibitors. Competitive chemoproteomic profiling of PF-9363 revealed this ligand dose-dependently engages MYST acetyltransferase complexes, but not other CoA-binding sites, in nuclear extracts. One caveat is that this approach exclusively reports on orthosteric, as opposed to allosteric, inhibition. This selective engagement allows clustering of chemoproteomic data and can be used to guide de novo identification of known and previously uncharacterized KAT complex members. Integrating chemoproteomic enrichment with AlphaFold screening was used to propose and validate an interaction between the FOXK2 and NSL complex. This chemoproteomic approach is complementary to existing approaches for protein-protein interaction discovery but is distinct from many in that it does not require a genetic tag. This raises the possibility of using it to analyze the composition of KAT complexes and native cells and tissues, and also potentially extending this approach to additional affinity capture matrices[51]. Unbiased LC-MS analysis of histone modifications recapitulated the hierarchical engagement of MYST acetyltransferase enzymes by PF-9363 as well as known KAT6 (H3K23Ac), KAT7 (H3K14Ac), KAT8 (H4K16Ac), and KAT5 (H2A.ZK4/K11Ac) biomarkers. An exception was H3K9Ac, which, in contrast to previous studies[8] we did not find to be sensitive to MYST inhibition. Given our results and the known regulation of H3K9Ac by KAT2A/B[52], we recommend caution when interpreting this modification as a biomarker of MYST KAT activity.

Target engagement data and cellular profiling studies converged on a panel of dose-responsive acetylation biomarkers for each MYST enzyme: H3K23Ac for KAT6A/B, H3K14Ac for KAT7, H4K16Ac for KAT8, and H2A.ZAc for KAT5. Evaluating these marks across multiple MYST inhibitors revealed their surprising properties. For example, PF-9363 and PF-8144 are the only MYST inhibitors that, at high concentrations, are able to inhibit KAT8-catalyzed H4K16Ac. WM-1119 appears to be a highly specific KAT6A/B inhibitor, leaving other biomarkers untouched even at high concentrations. WM-3835, marketed commercially as a KAT7 inhibitor, potently inhibits KAT6A/B-catalyzed H3K23Ac. These findings highlight the distinction between applying these compounds as chemical probes—in which they target a single enzyme, e.g., KAT6A/B—versus modulators, in which their pharmacology is leveraged to engage additional targets (e.g., KAT7/KAT8) at higher concentrations. As a general rule, we recommend monitoring these histone marks when deploying MYST inhibitors in novel systems and/or at escalated doses. This straightforward experiment, which can be carried out using commercial antibodies, provides a powerful measure to ensure that mechanistic effects are properly attributed to a given KAT enzyme's inhibition.

Evaluation of PF-9363 in the NCI-60 cell line panel indicated that acute inhibition of KAT6 and KAT7 ($\leq 1 \mu M$) is well tolerated, whereas growth inhibition at higher concentrations likely reflects KAT8 blockade. As an example, the triple-negative breast cancer line BT-549—which exhibits minimal dependency on EP300/CREBBP – was sensitive to high-dose PF-9363 ($GI_{50} \sim 7 \mu M$) that coincided with H3K23Ac, H3K14Ac, and H4K16Ac ablation. It has previously been suggested that cell lines with low KAT8 expression may be more vulnerable to KAT8 inhibition[50]. Our results provide a roadmap for testing this hypothesis by exploiting the differential pharmacology of PF-9363 versus WM-8014/WM-3835, the latter of which lacks activity against KAT8. Another question is whether the hierarchical target engagement phenomenon described here has clinical implications. The phase 1 clinical trial of PF-8144 in breast cancer provided data indicating modulation of the KAT6-regulated H3K23Ac[10]. Whether KAT7 or KAT8-regulated biomarkers are impacted by any of the MYST inhibitors currently undergoing clinical testing is unknown, but they should benefit from the thorough characterization reported here.

During preparation of this manuscript, a study by Perner et al. reported that dual KAT6/KAT7 inhibition using PF-9363 synergizes with Menin inhibitors to overcome primary and acquired drug resistance in MLL-rearranged leukemia[53]. This timely work illustrates how an understanding of dose-dependent target occupancy can be used to design strategies that rationally leverage MYST inhibitor polypharmacology for therapeutic benefit. By defining the dose-response relationships that govern small molecule engagement of MYST complexes, our studies provide a framework for interpreting the mechanism of action, probing MYST-dependent acetylation in diverse settings, and monitoring the selectivity of next-generation KAT inhibitors.

## Methods

### General materials and methods

Unless otherwise specified, chemicals and solvents were purchased from Sigma, VWR, or Fisher and used without further purification. PF-9363 (HY-132283), WM-1119 (HY-102058), WM-3835 (HY-134901), WM-8014 (HY-102060), PF-8144 (HY-153444), MOZ-IN-3 (HY-149470) and CPI-1612 (HY-136285) were purchased from MedChemExpress. MCF-7, BT-549 and HEK-293T cells were obtained from the NCI tumor cell repository. Their catalog numbers are as follows: MCF-7 (ATCC # HTB-22), HEK-293T (ATCC # CRL-3216), and BT-549 (ATC # HTB-122). Protein concentrations were determined using the Precision Red Protein Assay (Cytoskeleton, ADV02). Protein samples were prepared for SDS-PAGE by adding LDS buffer (Invitrogen, NP0007) containing 100 mM DTT, followed by denaturation at 95 °C for 10 min. Samples were loaded onto NuPAGE 4–12% Bis-Tris gels (Invitrogen) and electrophoresed at 160 volts for 1 h using XCell SureLock Mini-Cells (Invitrogen, EI0002) with MES running buffer (Invitrogen, #NP0002) according to the manufacturer's protocols. For detection of MYST proteins, gels were transferred to nitrocellulose membranes using the iBlot dry blotting system (Invitrogen, IB1001) with nitrocellulose transfer stacks (Invitrogen, IB301001) using program 0. For histone acetylation assessment, a modified transfer protocol was employed using the same instrument and transfer stacks but at 20 V for 8 min. Total protein on western blots was visualized with Ponceau staining, followed by two washes with 5% acetic acid in ddH$_2$O. Membranes were blocked with StartingBlock (PBS) Blocking Buffer (Thermo Scientific, 37538) for 30 min at room temperature and then incubated with primary antibodies at the indicated dilutions in StartingBlock Blocking Buffer overnight at 4 °C. The following primary antibodies were used: anti-KAT7 (Abcam, AB70183, 1:1000), anti-KAT8 (Cell Signaling, 46862S, 1:1000), anti-Naa50 (Proteintech, 16120-1-AP, 1:1000), anti-Lamin A/C (Bethyl Laboratories, A303-431A, 1:2000), anti-H3K23ac (Millipore, 07-355, 1:10000), anti-H3K14ac (Millipore, 07–353, 1:1000), anti-H4K16ac (Millipore, 07–329, 1:1000), acetyl-histone H2A.Z (Lys4/Lys7) (Cell Signaling, 75336, 1:1000), anti-H3K18ac (Millipore, 07–354, 1:1000), anti-H4 (Cell Signaling, 2935S, 1:1000), anti-H3K27me (Cell Signaling, 84932, 1:1000), anti-H3K27me2 (Cell Signaling, 9728, 1:1000), anti-H3K27me3 (Cell Signaling, 9733, 1:1000), anti-H3K79me (Cell Signaling, 12522, 1:1000), anti-H3K79me2 (Cell Signaling, 5427, 1:1000), anti-H3K79me3 (Cell Signaling, 74073, 1:1000), anti-Flag tag (Cell Signaling, 14793S, 1:1000) and anti-Myc tag (Cell Signaling, 2278S, 1:1000). Following primary antibody incubation, membranes were washed three times with 1× TBST (Cell Signaling, 9997) at room temperature and incubated for 1 h with either anti-rabbit IgG HRP-linked antibody (Cell Signaling Technology, 7074S) or anti-mouse IgG HRP-linked antibody (Cell Signaling Technology, 7076S), diluted 1:1000 in 1× TBST containing 5% non-fat dry milk. After three additional washes with 1× TBST, membranes were developed using either

Lumiglo (Cell Signaling Technology, #7003) or SuperSignal ELIZA Femto Substrate (Thermo Scientific, 37074) according to the manufacturer's protocols. Images were captured using an Amersham ImageQuant 800 imaging system (Cytiva, 29399482).

## KAT capture and competitive chemoproteomic profiling

**Preparation of H3K14-CoA affinity resin and capped beads.** The preparation of magnetic bead H3K14-CoA affinity resin was adapted from a previously reported protocol[54]. Briefly, 15.5 mg (1 eq) of a purified peptide with the linear sequence Ahx-QTARKSTGGK(BrAc) APRKQLATK-NH$_2$ (MW = 2260.48 g/mol; UNC High-Throughput Peptide Synthesis Facility) was added to a solution of Coenzyme A sodium salt hydrate (13.9 mg, 2.5 eq, Cayman, 21722) in 750 μL of 100 mM NaHCO$_3$. The reaction mixture was incubated at 37 °C with 300 rpm agitation for 2 h, then lyophilized. The dried product was reconstituted in 1 mL of 0.1% TFA in water and purified by HPLC using a 100-min gradient (buffer A: H$_2$O + 0.1% TFA; buffer B: acetonitrile, gradient 25–45% buffer B). The purified Ahx-H3K14-CoA peptide was characterized by LC-MS and lyophilized overnight. For bead functionalization, the Ahx-H3K14-CoA peptide was dissolved in PBS to a concentration of 1 mM. NHS-magnetic beads (550 μL; Pierce, 88827, 10 mg/mL) were pre-washed twice with cold PBS and resuspended in cold PBS to maintain a concentration of 10 mg/mL. The peptide-CoA solution (500 μL, 1 mM) was added to the bead suspension, and the pH was adjusted above 8.0 by adding 12 μL triethylamine. The mixture was rotated at 4 °C overnight. The beads were then magnetically separated for 3 min, the supernatant was removed, and the beads were washed once with 1 mL H$_2$O. To block unreacted NHS groups, the functionalized beads were incubated with 1 mL of 1 M ethanolamine solution (pH 8.3) using end-over-end mixing for 3 h at room temperature. The beads were washed three times with wash buffer (100 mM HEPES, pH 7.5, 500 mM NaCl, 1 mL per wash), followed by four washes with IP buffer (50 mM HEPES, pH 7.5, 150 mM NaCl, 0.1% NP-40, 1 mL per wash). The beads were finally resuspended in 500 μL IP buffer to yield a 10 mg/mL suspension and stored at 4 °C. Control capped beads were prepared using an identical protocol, except that PBS was substituted for the peptide-CoA solution during the bead coating step.

**Preparation of KAT capture samples for analysis by immunoblotting & LC-MS/MS.** KAT capture and competitive capture using H3K14-CoA magnetic beads, as well as sample preparation for immunoblotting, were performed similarly to previously reported protocols[54–56]. Briefly, HeLa nuclear extracts (IPRACELL, CC-01-20-50) were diluted to 1 mg/mL in high salt buffer (1× PBS [Quality Biological, 114-056-101], 200 mM NaCl, 0.1% NP-40, and protease inhibitor cocktail [Cell Signaling Technology, 5871]). For each assay, 400 μL of diluted nuclear extract was centrifuged (20,000 × g, 4 °C, 30 min) to remove protein precipitates, and the clarified supernatant was transferred to a fresh microcentrifuge tube. Nuclear extracts were pre-incubated with varying concentrations of PF-9363 (0, 0.01, 0.1, 1, 10, or 30 μM) by adding 2 μL of 200× stock solutions in DMSO, followed by rotation at 4 °C for 2 h with end-over-end mixing. After inhibitor pre-treatment, 20 μL of H3K14-CoA magnetic beads was added to each sample, and the mixtures were rotated for an additional 2 h at 4 °C. Beads were collected using a magnetic rack, supernatants discarded, and the resin was subjected to a series of mild washes (1 × 400 μL high salt buffer, followed by 2 × 400 μL standard wash buffer [50 mM HEPES pH 7.5, 150 mM NaCl]). Following the final wash, bound proteins were eluted by resuspending beads in 50 μL of 1× LDS buffer containing 100 mM DTT and heating at 95 °C for 10 min. After magnetic separation for 3 min, the eluted material was transferred to a fresh tube. A second elution was performed using the same conditions, and both eluates were combined. For immunoblot analysis, 20 μL of each elution sample was resolved on NuPAGE 4-12% Bis-Tris gels at 160 V for 1 h and

transferred to nitrocellulose membranes using the iBlot dry blotting system (program 0). Membranes were probed with antibodies against KAT8, KAT7, Naa50, and Lamin A/C to evaluate the dose-dependent competition effects of PF-9363 on protein binding to the H3K14-CoA affinity resin.

Preparation of samples for LC-MS/MS analysis was scaled up 2.5-fold and performed analogously with the following specific modifications: three biological replicates were performed for each treatment condition to ensure statistical robustness. Control experiments using capped beads were processed identically to assess non-specific binding interactions: 1000 μL of 1 mg/mL nuclear extract per sample, a modified inhibitor concentration series (0, 0.1, 1, and 10 μM) for PF-9363 and WM-3835, 50 μL of H3K14-CoA beads per sample, proportionally increased wash volumes (1 mL per wash). Instead of LDS buffer elution, beads were resuspended in 150 μL of a solution containing 50% Easypep lysis buffer, 25% reducing solution, 25% alkylating solution, and 8 ng/μL trypsin/LysC (all from Thermo Scientific, A40006) and incubated at 37 °C for 24 h. From each sample 180 μL was removed from the beads and treated with 20 μL of 5 μg/μL TMTpro and incubated at 25 °C for 1 h, then quenched with 50 μL of 5% Hydroxylamine, 20% formic acid for 10 min and combined. Combined TMTpro samples were cleaned using EasyPep mini columns (Thermo Scientific, A40006) and eluted in 300 μL and split into 2 × 150 μL aliquots and dried. For the KAT6A spike-in experiment, 5 μg of KAT6A (N-term Flag-tagged full-length hsKAT6A from Pfizer) was spiked into 1 mg of clarified HeLa nuclear extracts (1 mg/mL) for per sample. 5 μL of DMSO, 200 μM PF-9363 or 200 μM PF-8144 was added to KAT6A spike-in proteome, and pre-incubated for 2 h at 4 °C by rotation. After inhibitor pre-treatment, 100 μL of H3K14-CoA magnetic beads was added to each sample and incubated for an additional 2 h at 4 °C by end-over-end mixing. Control experiments using capped beads were processed identically to assess non-specific binding interactions. Three times wash was performed, followed by trypsin digestion and TMT labeling, which were the same as the KAT capture sample preparation mentioned before. Three biological replicates were performed for each condition.

**TMT-based LC-MS/MS analysis and data processing.** Dried peptides from one aliquot were resuspended in 50 μL of 0.1% FA, and 10 μL was analyzed in duplicate using a Dionex U3000 RSLC in front of an Orbitrap Eclipse (Thermo Scientific) equipped with an EasySpray ion source. Solvent A consisted of 0.1%FA in water, and Solvent B consisted of 0.1%FA in 80%ACN. Loading pump consisted of Solvent A and was operated at 7 μL/min for the first 6 min of the run then dropped to 2 μL/min when the valve was switched to bring the trap column (Acclaim™ PepMap™ 100 C18 HPLC Column, 3 μm, 75 μm I.D., 2 cm, PN 164535) in-line with the analytical column (EasySpray 75 μm I.D., 25 cm, PN ES902). The gradient pump was operated at a flow rate of 300 nL/min, and each run used a linear LC gradient of 5–7%B for 1 min, 7–30%B for 84 min, 30–50%B for 25 min, 50–95%B for 4 min, holding at 95%B for 7 min, then re-equilibration of analytical column at 5%B for 17 min. All MS injections employed the TopSpeed method with three FAIMS compensation voltages (CVs) and a 1 s cycle time for each CV (3 s cycle time total) that consisted of the following: Spray voltage was 2200 V and ion transfer temperature of 300 °C. MS1 scans were acquired in the Orbitrap with resolution of 120,000, AGC of 4e5 ions, and max injection time of 50 ms, mass range of 375–1600 m/z; MS2 scans were acquired in the Orbitrap using TurboTMT method with resolution of 15,000, AGC of 1.25e5, max injection time of 22 ms, HCD energy of 38%, isolation width of 0.4 Da, intensity threshold of 2.5e4 and charges 2–5 for MS2 selection. Advanced Peak Determination, Monoisotopic Precursor selection (MIPS), and EASY-IC for internal calibration were enabled, and dynamic exclusion was set to a count of 1 for 15 s. The only difference in the methods was the CVs used, one method used CVs of −45, −60, −75 and the second used CVs of −55, −70, −85. Raw MS

files were searched with Proteome Discoverer 2.4 using the Sequest node. Data was searched against the Uniprot Human database from August 2023 using a full tryptic digest, 2 max missed cleavages, minimum peptide length of 6 amino acids and maximum peptide length of 40 amino acids, an MS1 mass tolerance of 10 ppm, MS2 mass tolerance of 0.02 Da, variable oxidation on methionine (+15.995), fixed TMTpro (+304.207) on lysine and peptide N-terminus, and fixed cysteine modification of carbamidomethyl (+57.021). Percolator was used for FDR analysis, and TMTpro reporter ions were quantified using the Reporter Ion Quantifier node and normalized on the total peptide intensity of each channel. TMTpro channel assignment for conditions can be found in Supplemental Data. Only proteins with high confidence were selected at 1% FDR, and a two-tailed t-test was performed to generate volcano plots. Raw mass spectrometry proteomics files and database search results have been deposited at the ProteomeXchange Consortium (http://proteomecentral.proteomexchange.org) with data set identifier PXD074488.

### Deep global profiling of HeLa nuclear extracts

**Sample preparation for deep global profiling.** 80 μg of HeLa nuclear extracts (IPRACELL, CC-01-20-50) were prepared in triplicate and diluted to 1 mg/mL in high salt buffer (1× PBS [Quality Biological, 114-056-101], 200 mM NaCl, 0.1% NP-40, and protease inhibitor cocktail [Cell Signaling Technology, 5871]), followed by alkylation and digestion in 200 μL of 1:1:1:1 100 mM HEPES pH 8/EasyPep Lysis buffer/reducing agent/alkylating agent from EasyPepTM MS Sample Prep Kit (Thermo Scientific, A40006) containing 5 ug of trypsin-LysC mix (Thermo Scientific, A40007) for overnight at 37 °C with shaking at 600 rpm. The samples were cleaned up using the EasyPepTM Mini 96-well plate format (Thermo Scientific, A45733) by following the manufacturer's protocol. The collected peptides were dried under speed-vac and redissolved in 50 μL of 0.1% formic acid in water. Experiments were performed in triplicate. The peptides were then subjected to an offline high pH reversed-phase fraction using a Waters Acquity UPLC system connected to an Xbridge Peptide BEMTM 2.5 μm C18 column (150 mm × 3.0 mm, Waters) running at 0.35 mL/min with detection using a fluorescence spectrometer (Waters). The peptides were separated using mobile phase solvent A (10 mM ammonium formate, pH 9.4) and mobile phase solvent B (90% acetonitrile in 10 mM ammonium formate, pH 9.4). with the following gradients: 6% solvent B (0–1 min), 6–10% solvent B (1–1.5 min), 10–50% solvent B (1.5–60 min), 50–90% solvent B (60–65 min), 90–6% solvent B (65–70 min). Fractions were collected in 350 μL volumes for a total of 60 fractions and were pooled into 6 fractions by concatenation. The fractions were dried under speed-vac, and the peptides were dissolved in 3% acetonitrile + 0.1% formic acid in water.

**Label-free LC-MS/MS analysis and data processing.** Each of the peptide samples was loaded onto the same LC-MS system (Dionex and Orbitrap Eclipse as described above) with the same column specifications and loading pump settings. The gradient pump was set to a flow rate of 300 nL/min and used a linear LC gradient of 5–7% B for 1 min, 7–30% B for 134 min, 30–50% B for 35 min, 50–95% B for 4 min, holding at 95% B for 7 min, then re-equilibration to 5% B for 17 min. All runs used the TopSpeed method with 3 s cycle time that consisted of spray voltage at 2200 V and ion transfer temperature of 275 °C. MS1 scans were acquired in the Orbitrap with a 120,000 resolution, AGC target of 4e5 ions, max injection time of 50 ms, and mass range set to 400–1600 m/z; MS2 scans were acquired in the Orbitrap with a resolution of 15,000, standard AGC target of 5e4 ions, max injection time of 22 ms, HCD energy of 30%, isolation window of 1.6 Da, minimum intensity set at 2.5e4, and charges 2–6 for MS2 selection. Advanced Peak Determination, Monoisotopic Precursor Selection (MIPS), and EASY-IC for internal calibration were all enabled with dynamic exclusion set to a count of 1 for 15 s. The raw files were searched against the human proteome database (UP000005640) from Uniprot (Accessed in November 2023) using Andromeda embedded in MaxQuant (version 2.4.13.0). A full tryptic digestion was assumed with allowed missed cleavages up to 2. Fixed modifications were set for cysteine carbamidomethylation (+57.021) and variable modifications for methionine oxidation (+15.995) and protein N-term acetylation (+42.011). The intensity-based absolute quantification (iBAQ) values were extracted for each protein. The data was filtered on Perseus (version 2.0.11) by removing proteins only identified by site, reversed, or as a contaminant. The iBAQ values were log-transformed and replotted on Microsoft Excel. Raw mass spectrometry proteomics files and database search results have been deposited at the ProteomeXchange Consortium (http://proteomecentral.proteomexchange.org) with data set identifier PXD074488.

### Identification of candidate MYST complex members from chemoproteomic competition data

**Dimensionality reduction and clustering.** Proteomics data containing gene identifiers, log2 fold change values, and associated p-values across three experimental conditions (0.1 v. 0 μM PF-9363, 1 v. 0 μM PF-9363, and 10 v. 0 μM PF-9363) were processed using a custom Python script. Only gene entries with nominal p-values less than 0.05 in at least one condition were retained, ensuring clustering focused exclusively on statistically significant changes. We used $\log_2$ fold change values as primary features for clustering analysis. Prior to analysis, entries with missing values were excluded, and features were normalized to zero mean and unit variance. Dimensionality reduction was achieved using t-distributed Stochastic Neighbor Embedding (t-SNE) with perplexity dynamically adjusted based on dataset size (maximum 30) and using two output dimensions. Unsupervised k-means clustering was subsequently applied with the number of clusters set to five. To distinguish between tightly grouped and diffuse clusters, compactness was quantified by calculating the average pairwise Euclidean distance between all points within each cluster. Clusters with lower average pairwise distances were considered more compact, suggesting stronger biological relationships. Results were visualized as two-dimensional scatter plots with points colored according to cluster assignments, revealing distinct gene expression patterns across treatment conditions.

**Structure prediction.** Binding structures between FOXK2 (Uniprot: Q01167) and candidate interactors identified from our chemoproteomic data were predicted using AlphaPulldown[33] v0.30.7.4 with multiple sequence alignments generated via ColabFold Search v1.5.5 on the NIH HPC Biowulf Cluster. Five models were generated per protein-protein pair without using templates or paired MSAs. Local Interaction Scores (LIS) and Local Interaction Area (LIA) were calculated according to previously described formulas[57]. Structural visualizations were generated using ChimeraX v1.8. The results generated in this study have been deposited in Zenodo (https://zenodo.org/records/15238967)[58].

**Composite scoring of candidate FOXK2-protein interactions.** To prioritize potential interactions, we developed a composite scoring system integrating multiple AlphaFold metrics (mpDockQ/pDockQ, LIS, and LIA), building upon established approaches for protein-protein interaction analysis[59]. Each interaction was evaluated against minimum acceptable thresholds (1610 for LIA, 0.073 for LIS, and 0.175 for mpDockQ/pDockQ) and assigned a weighted constant (k): 1.0 for passing all three thresholds, 0.75 for two, 0.5 for one, and zero for none. Metrics were then min-max normalized, summed, and multiplied by the weighted constant to generate final composite scores for ranking candidate interactions. All related data, code, and analysis scripts have been deposited to Zenodo (https://zenodo.org/records/15238967)[58].

**Prediction of interface contacts.** The two highest-ranking interactors by composite score, OGT (Uniprot: O15294) and WDR5 (Uniprot: P61964), were further analyzed to identify interface residue-residue contacts using both AlphaFold2 and AlphaFold3. For AlphaFold2 analysis[60] we calculated Euclidean distances between alpha carbon coordinates of residues using Sci-Py's Distance Matrix Module, filtering for interactions under 8 Å. Highly interacting residues were defined as those with distances less than 6 Å and PAE scores less than 25.7. For AlphaFold3 analysis[61], we extracted the contact_probs metric from the summary JSON file, with highly interacting areas identified by contact probability values greater than 0.1. Predictions from both methods were compared to identify overlapping interface residues, and structural visualizations were generated using ChimeraX v1.8. All data, code, and analysis scripts have been deposited to Zenodo (https://zenodo.org/records/15238967)[58].

### Ectopic expression and co-immunoprecipitation of FOXK2 interactors

**Cloning.** Plasmid constructs containing the full-length of FOXK2 with a C-terminal Flag tag, or full-length of WDR5 with a C-terminal Myc tag, or full-length of OGT with an N-terminal Myc tag were ordered from TWIST, and pTwist CMV Puro backbone was used for all the plasmids. They all validated through Sanger and whole-plasmid sequencing. The inserted sequences were shown in Supplementary Data 9.

**Ectopic expression and co-immunoprecipitation.** HEK-293 cells were plated in 60 mm dishes (4.4*10^5 cells/dish in 4.4 ml DMEM media). After 24 h, plasmids were transfected using Fugene 6 (Promega, E2691) according to the manufacturer's instructions: 202.8 μL of OPTIM, 17.2 μL of Fugene 6 and 2.3 μg of plasmid in total were used per dish. 1.15 μg of FOXK2-Flag tag plasmid with 1.15 μg of WDR5-Myc or Myc-OGT plasmid were used for co-overexpression. Overexpression/Co-overexpression was carried out by incubating the cells for 48 h at 37 °C under 5% CO2 atmosphere, after which the cells were harvested. For basal conditions, 500 μL of lysis buffer was added to each dish, followed by 30 min incubation at 4 °C. Lysates were clarified by centrifuge (4 °C, 10 min at 12,000 g), then quantified by Precision Red and normalized. Anti-FLAG pulldown was performed using the FLAG Immunoprecipitation Kit (Millipore, FLAGIPT1) according to the manufacturer's instructions. 0.7 mg of the lysate was incubated with 20 μL of pre-washed anti-FLAG gel slurry overnight at 4 °C. Purified protein was run on SDS-PAGE and immunoblotted against anti-Myc-tag, anti-FLAG-tag, anti-KAT8 and anti-Naa50 as described in General Materials and Methods. For the high salt buffer wash condition, cells were resuspended in high salt buffer and then sonicated at 25% amplitude, 3 s pulse, 12 s off, for 3 cycles on ice. Lysates were centrifuged (4 °C, 10 min at 12,000 g), followed by quantification by Precision Red and normalization. Anti-FLAG pulldown was processed using the same FLAG Immunoprecipitation Kit (Millipore, FLAGIPT1) according to the manufacturer's instructions. Only difference from basal condition is the wash step after incubation: for high salt buffer wash groups, 1 mL of high salt buffer was used for 1st time wash. And 1 mL of wash 2 buffer (50 mM HEPES pH 7.5, 150 mM NaCl) was used for the 2nd and 3rd time wash.

### Histone modification analysis

**Cell culture.** MCF-7 cells were cultured at 37 °C under 5% CO$_2$ in EMEM (Quality Biological, 112-016-101) with 0.01 mg/mL human recombinant insulin (Sigma Aldrich, 91077 C), 2 mM L-glutamine (Quality Biological, 118-084-721), and 10% FBS (Avantor Seradigm, 97068-085). BT-549 cells were cultured at 37 °C under 5% CO2 in RPMI-1640 (Quality Biological, 112-024-101) with 10% FBS, 2 mM L-glutamine, and 0.023 U/mL human recombinant insulin.

**Analysis of histone modifications by immunoblotting.** MCF-7 cells (3.8 × 105) or BT-549 cells (3.5 × 105) were plated per well in 6-well dishes. Experiments were performed in duplicate. After 24 h recovery, the media was replaced with 2 mL of inhibitor-containing media to yield final concentrations of 0, 0.1, 1, 10, or 30 μM of the specified inhibitor (PF-9363, WM-8014, WM-1119, WM-3835, PF-8144, MOZ-IN-3). Following 24 h inhibitor treatment, cells were washed with ice-cold 1× PBS and scraped in 150 μL of Nuclear Isolation Buffer (NIB) containing 0.1% NP-40 for histone extraction using a modified protocol from the Garcia laboratory[62]. Briefly, cell suspensions were incubated on ice for 5 min, and nuclei were pelleted by centrifugation (600 g, 4 °C, 5 min). Pelleted nuclei were washed twice with 150 μL of detergent-free NIB buffer, followed by centrifugation (600 g, 4 °C, 5 min) to remove all detergent. Nuclei were then resuspended in 400 μL of 0.4 N H$_2$SO$_4$ and rotated at 4 °C overnight for acid extraction of histones. The following day, samples were centrifuged (11,000 g, 4 °C, 10 min) and the histone-containing supernatant was transferred to a fresh tube. Trichloroacetic acid (100% TCA) was added to achieve a final concentration of 20% (v/v), tubes were inverted once to mix, and histone precipitation was performed on ice for at least 3 h or overnight at 4 °C. Precipitated histones were collected by centrifugation (11,000 g, 4 °C, 10 min) and the supernatant carefully removed, leaving a visible film of histones on the tube wall or bottom. Histone pellets were subjected to sequential washes with 1 mL of ice-cold acidified acetone (0.1% 12 N HCl) followed by 1 mL of 100% ice-cold acetone, with centrifugation (11,000 g, 4 °C, 5 min) after each wash. Samples were air-dried at room temperature, resuspended in 40 μL of ddH$_2$O, and protein concentration determined using the Precision Red Protein Assay (Cytoskeleton, ADV02). For immunoblot analysis, 1–2 μg of purified histones were loaded per lane and processed according to the general immunoblotting protocol described in General Materials and Methods.

**Analysis of histone modifications by LC-MS/MS.** Preparation of histone samples for LC-MS/MS was performed analogously to the procedure for immunoblotting above, with the following modifications. MCF7 cells (4.2 × 10$^6$) were seeded in 15 cm dishes and treated with PF-9363 or vehicle. Experiments were performed in triplicate. Following treatment and PBS washing, cells were scraped and collected in 15 mL Falcon tubes, pelleted (500 g, 4 °C, 5 min), resuspended in 1 mL cold PBS, transferred to microcentrifuge tubes, and pelleted again before flash-freezing in liquid nitrogen for storage at −80 °C. For extraction, cell pellets (~100 μL) were resuspended in NIB buffer at a 1:10 ratio, and cells were lysed with NIB buffer containing 0.2% NP-40 for 8 min on ice. Nuclei were pelleted at 1000 g for 8 min and washed three times with 500 μL detergent-free NIB buffer. Histones were extracted using 800 μL of 0.2 M H$_2$SO$_4$, and after centrifugation (3,400 g, 4 °C, 10 min), supernatants were further clarified at 5000 g before TCA precipitation (33% final concentration). Histone pellets were collected (5000 g, 4 °C, 10 min), washed at 7000 g, and resuspended in 100 μL ddH$_2$O. Sample quality was verified by SDS-PAGE with Coomassie Blue staining prior to bottom-up LC-MS analysis[62]. Briefly, histone lysine residues were derivatized using a propionylation reagent (regent-to-sample ratio of 1:2) composed of acetonitrile and propionic anhydride in a 3:1 ratio. The pH of the solution was adjusted to 8.0 using ammonium hydroxide. Propionylation was performed twice, with a 15-min incubation at room temperature between each round. The samples were then dried on a SpeedVac. The derivatized histones were digested with trypsin at a 1:50 ratio (wt/wt) in 50 mM ammonium bicarbonate buffer at room temperature overnight. Subsequently, the N-termini of the resulting histone peptides were subjected to two rounds of propionylation and dried again using SpeedVac[63]. Peptides were desalted using self-packed C18 stage tips, then dried and reconstituted in 0.1% formic acid. Peptide analysis was conducted using a Vanquish Neo UHPLC coupled to an Orbitrap

Exploris 240 mass spectrometer (Thermo Scientific). Samples were maintained at 7 °C in the LC autosampler. Peptide separation was achieved using an Easy-Spray™ PepMap™ Neo nano-column (2 μm, C18, 75 μm × 150 mm) at room temperature. The LC gradient consisted of a linear increase from 2% to 32% solvent B (0.1% formic acid in acetonitrile) in solvent A (0.1% formic acid in water) over 48 min, followed by an increase to 98% solvent B over the next 12 min, at a flow rate of 300 nL/min. Mass spectrometry was performed using data-independent acquisition (DIA) mode. Each acquisition cycle included a full MS scan followed by 35 DIA MS/MS scans with 24 m/z isolation windows spanning from 295 to 1100 m/z. Full MS scans were acquired in the Orbitrap mass analyzer across 290–1100 m/z at a resolution of 60,000, in positive profile mode, using an auto maximum injection time and an AGC target of 300%. MS/MS data from HCD fragmentation were collected in the Orbitrap. These scans typically used an NCE of 30, an AGC target of 1000%, and a maximum injection time of 60 ms. Histone peptide data were analyzed using EpiProfile 2.0[64]

### Cytotoxicity assays

NCI-60 cell lines were treated with PF-9363 or EP300/CREBBP inhibitor CPI-1612 at five concentrations (0.01, 0.1, 1, 10, 100 μM) for 48 h and analyzed by sulforhodamine B assay for growth inhibition as previously described[65–67]. To validate the NCI-60 relative activity trends using an orthogonal method, we assessed PF-9363 activity against BT-549 cells using an ATP-based viability assay. Briefly, BT-549 cells ($3 \times 10^3$ per well) were seeded in white-walled 96-well plates (Corning 3610) and allowed to recover for 24 h. Cells were then treated with a 10-point dilution series of PF-9363 (60, 30, 15, 7.5, 3.75, 1.88, 0.94, 0.47, 0.23, 0 μM) for 96 h. Experiments were performed in quadruplicate, and cell viability was measured using CellTiter-Glo® Luminescent Cell Viability Assay (Promega, G7572) according to the manufacturer's instructions. Luminescence signals were recorded on a BioTek Synergy 2 plate reader, and results were normalized to vehicle-treated controls (set to 100%). Half-maximal inhibition values ($IC_{50}$) were determined from nonlinear regression analysis of dose-response curves using GraphPad Prism 9.

### Reporting summary

Further information on research design is available in the Nature Portfolio Reporting Summary linked to this article.

## Data availability

The data supporting the findings of this study are available within the article and its Supplementary Figures. Structure prediction and interface contacts prediction data generated in this study have been deposited to Zenodo [https://zenodo.org/records/15238967][58]. Raw mass spectrometry proteomics files and database search results have been deposited at the ProteomeXchange Consortium [http://proteomecentral.proteomexchange.org] with the dataset identifier PXD074488. The accession code of the crystal structure of PF-9363 bound to KAT6A catalytic domain is: 8DD5. Other data generated in this study are provided in the Supplementary Information/Supplementary Data /Source Data file. Source data are provided with this paper.

## Code availability

The code used to predict structure/interface contacts and assess FOXK2 interactors in this study has been deposited to Zenodo [https://zenodo.org/records/15238967][58]

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

## Acknowledgments

The authors thank Whitney Lieberman (NCI), Diamond Gallimore (NCI), and Krzysztof Krajewski (UNC) for assistance with materials and pilot experiments, and Nathan Coussens (DTP) for NCI-60 screening, Oleg Brodsky (Pfizer) for the kind gift of KAT6A, and Francis O'Reilly (NCI) and Thomas Paul (Pfizer) for helpful discussions. Portions of Figs. 2A, 3D, and 4A were generated using image templates from BioRender.com under the institutional license belonging to the National Cancer Institute. This work was supported by the Intramural Research Programs of the

National Cancer Institute, Center for Cancer Research ZIA BC011488 (J.L.M.) St. Jude Children's Research Hospital/P22-07377 (B.A.G.) and NIH/P01-CA196539-11 (B.A.G.). This work utilized the computational resources of the NIH HPC Biowulf cluster (http://hpc.nih.gov). In addition, this project has been funded in part with federal funds from the National Cancer Institute, National Institutes of Health, under contract numbers HHSN261200800001E and HHSN261201500003I. The content of this publication does not necessarily reflect the views or policies of the Department of Health and Human Services, nor does mention of trade names, commercial products, or organizations imply endorsement by the U.S. Government.

## Author contributions

Conceptualization: X.C., A.C., and J.L.M. Methodology: X.C., A.C., M.P., R.H., K.S., R.K. Software: A.C. Validation: X.C., A.C., M.P., R.H., K.S., T.A., and R.K. Formal analysis: X.C., A.C., R.H., K.S., T.A., R.K., and J.L.M. Investigation: X.C., A.C., M.P., R.H., K.S., and R.K. Resources: X.C., A.C., M.P., R.H., K.S., R.K. Data Curation: X.C., A.C., R.H., K.S., R.K., and J.L.M. Writing - Original Draft: X.C. and J.L.M. Writing - Review & Editing: X.C., A.C., M.P., R.H., K.S., R.K., B.G., and J.L.M. Visualization: X.C. and A.C. Supervision: J.L.M. and B.G. Project Administration: J.L.M. and B.G. Funding Acquisition: J.L.M.

## Competing interests

The laboratory of J.L.M. receives funding from Pfizer under a Collaborative Research and Development Agreement. The remaining authors declare no competing interests.
