## [Transparent Peer Review file · Nature Communications]

Hierarchical small molecule inhibition of MYST acetyltransferases

Corresponding Author: Dr Jordan Meier

Version 0:

Reviewer comments:

Reviewer #2

(Remarks to the Author)

The manuscript titled "Hierarchical Small Molecule Inhibition of MYST Acetyltransferases" highlights the therapeutic potential of MYST lysine acetyltransferases (KATs) as targets in cancer treatment. Several drug-like inhibitors, including PF-9363, WM-8014, WM-1119, and WM-3835, have demonstrated promising activity in preclinical and clinical settings. This study employs chemoproteomic profiling in conjunction with histone acetylation biomarkers to systematically assess the selectivity and potency of these MYST KAT inhibitors. The authors report a hierarchical inhibition pattern (KAT6A/B > KAT7 >> KAT8 > KAT5) and further identify novel components of the MYST complexes, such as FOXK2. Additionally, screening across the NCI-60 cell line panel reveals that PF-9363 can suppress the proliferation of epigenetic inhibitor-resistant cancer cells through KAT8 inhibition at higher concentrations. However, following are my critical concern:

1. According to the chemoproteomic analysis conducted to assess MYST inhibitor selectivity, KAT6A and KAT6B were not detected in the sample output. In contrast, biochemical assays identified KAT6A and KAT6B as the most potent targets of PF-9363. This discrepancy highlights a significant limitation—or "blind spot"—of the chemoproteomic approach. The study proposes a hierarchical inhibition profile for the KAT family in the order: KAT6A/B > KAT7 >> KAT8 > KAT5. While a brief explanation for this discrepancy is provided, further validation is necessary, particularly to investigate the low abundance of KAT6A/B and their inefficient extraction from the chromatin fraction.
2. Chemoproteomic analysis works on active site competition. How will the allosteric side competitors be addressed?
3. The quality and exposure of the H4 western blot presented in Figure 4 could be improved to enhance clarity and interpretability.
4. A better explanation is recommended on how FOXK2 is relevant for this paper and AN IP experiment to validate the in Silico data.
5. To enable a direct comparison of the dose-dependent effects of the inhibitors on MYST-dependent biomarkers, MCF-7 cells were treated with varying concentrations. To validate the specificity of these effects, parallel treatment of MYST knockdown cells with the inhibitors is recommended.
6. A clearer explanation is recommended to clarify the relevance of FOXK2 in the context of this study. Additionally, an immunoprecipitation (IP) experiment would be valuable to experimentally validate the in silico findings.
7. Reframe the sentence: Target engagement was assessed using the panel of MYST-dependent histone acetylation biomarkers defined above. Analyzing the first-generation sulfonohydrazide WM-8014, we observed it was old less potent than PF-9363 but afforded similarly complete inhibition of KAT6A/B – as assessed by inhibition of H3K23Ac – at 1 μ M (Fig. 5B)
8. Typing error - Our chemoproteomic assay cannot distinguish whether this reflects preferential capture of the BRPF complex by the affinity resin, preferential engagement of the the JADE complex by PF-9363, or association of BRPF3 with another MYST complex.
9. The dose-dependent inhibition (e.g., 10–30 μ M) of KAT7/KAT8/KAT5 may not be pharmacologically relevant.
10. The author can use consistent naming for KATs throughout (either KAT6A/B or KAT6A/KAT6B). (Introduction, Line 24)

(Remarks on code availability)

Reviewer #3

(Remarks to the Author)

In this study, Chen et al., used a chemoproteomics approach with immobilized CoA-resin and increasing concentrations of the MYST inhibitor PF-9363 to identify candidate interacting proteins in MYST complexes. They identified that different MYST complexes have different sensitivity to the inhibitor and this could be read functionally by decreased acetylation at specific histone residues known to be targets of those complexes. Finally, they tested how different cell lines were acutely sensitive to the inhibitors, revealing that some commercial inhibitors may not be marketed for the correct specificity. Overall I don't have concerns with the validity of the data, but rather, that the narrative and motivation for the study are unclear.

1. The paper would benefit from being written to be more accessibly so that non-experts can understand the work better. For example, if the authors want the wider scientific community to heed their warnings about the commercial inhibitors specificities
2. The authors report that the chemoproteomics approach did not identify KAT6 but that this is because KAT6 is low/not expressed in HeLa cells. If KAT6 were expressed, would this change the IC50 concentrations at which PF-9363 might inhibit other KATs?
3. The authors claim that their approach demonstrates a strategy for identifying new candidate interactions partners of multiprotein KAT complexes. Some additional discussion about why their approach might be better than other existing strategies people use to unbiasedly investigate novel protein-protein interactions in multiprotein complexes would be helpful. It seems like if the authors want to argue this way is better, they might need to actual validate some aspect of their AI model of the interactions?
4. The authors perform the MCF-7 experiments in Figure 4 to establish the physiological relevance of MYST complex inhibition. Could the authors comment on the justification or physiological relevance of the PF-9363 concentrations that they used? Is this effectively a similar experiment to Figure 2 except cells were treated directly so it's a functional readout of Kac peptides rather than protein complexes?
5. I understand for the NCI-60 cells in Figure 6 why one might want to broaden their inhibitor studies to validate them in more cells, but the reasoning that the long-term growth changes are known but not the short-term effects feels a little underdeveloped. Can the authors expand on why the short-term effects are important too, what new insights these experiments provide, and where we go from here?

(Remarks on code availability)

Reviewer #4

(Remarks to the Author)

This manuscript presents the development of a strategy to compare the selectivity and potency of several MYST inhibitors. The authors employ competitive chemoproteomics to evaluate the ability of PF-9363, a known KAT6A/6B inhibitor, to engage native MYST family complexes, revealing a hierarchical selectivity profile: KAT7 > KAT8 > KAT5 and their associated complex members. Further analysis of the chemoproteomic data led to the observation that depletion of FOXK2, a forkhead box transcription factor, mirrored that of MYST complex components. Subsequent AlphaPulldown analysis suggested that FOXK2 may interact with MYST complexes via WDR5.

Using bottom-up proteomics, the authors mapped dose-dependent loss of histone acetylation in PF-9363-treated cells to specific KATs. This approach allowed them to validate the compound's selectivity hierarchy—KAT6A/6B > KAT7 > KAT8—and to identify H3K23Ac, H3K14Ac, and H4K16Ac as biomarkers of KAT6A/B, KAT7, and KAT8 inhibition, respectively. These biomarkers were then used to assess the selectivity of sulfonylhydrazine-based inhibitors WM-8014, WM-1119, and WM-3835. Interestingly, WM-3835—marketed as a KAT7 inhibitor—appears to preferentially engage KAT6A/B.

Finally, the authors demonstrate that PF-9363 kills KAT8-dependent BT-549 cells using their functional biomarkers. Overall, this is a strong manuscript with clear therapeutic implications. The authors are transparent about the limitations of their approach and appropriately measured in their conclusions. I recommend it for publication, provided the following concerns are addressed:

Line 24 of the abstract contains a typo: "histone acetylation biomarkers" should be corrected.

Figure 2D: If feasible, inclusion of a KAT5 immunoblot would help corroborate the chemoproteomics results.

Pulldown assay limitations: The authors acknowledge the failure to recover KAT6A/B but do not experimentally address this limitation. Could alternative fractionation protocols or overexpression of KAT6A/B be tested?

Figure 3E and 3F are labeled "AF2 multimer prediction...", while the legend refers to "AF3-Multimer prediction...". This inconsistency should be corrected.

FOXK2 association: Dose-dependent depletion of FOXK2 upon PF-9363 treatment implies a potential association with MYST complexes, possibly through WDR5 or OGT. The authors should consider confirming this interaction via co-immunoprecipitation.

Line 40 of the section "Assessing target engagement by orthogonal MYST inhibitor chemotypes" uses the phrase "old less potent..."—this wording is awkward and should be revised for clarity.

(Remarks on code availability)

Version 1:

Reviewer comments:

Reviewer #2

(Remarks to the Author)

The authors have addressed all my concerns satisfactorily.

(Remarks on code availability)

Reviewer #3

(Remarks to the Author)

The authors have sufficiently addressed my comments.

(Remarks on code availability)

Reviewer #4

(Remarks to the Author)

The authors have adequately addressed my questions and concerns, and the manuscript has now become suitable for publication in Nature Communications.

(Remarks on code availability)

Enclosed is a revised version of “Hierarchical small molecule inhibition of MYST acetyltransferases” to be considered for publication in *Nature Communications*. We were encouraged by the reviewer comments, and thank them for their thoughtful feedback. All three referees recognized the importance of the subject area, its therapeutic implications, and the value of a resource defining the properties of MYST inhibitors for the epigenetic drug discovery field. The reviews also highlighted experiments that could strengthen the paper – most notably by 1) confirming the newly discovered FOXK2/NSL interaction, 2) demonstrating full consistency between chemoproteomic and biomarker analyses of inhibitor specificity, and 3) more clearly stating the goals and significance of our study. These critiques were greatly appreciated and prompted significant additional experimentation as well as revisions to the text. Overall, these changes address the reviewer’s concerns and have greatly improved the manuscript.

Reviewer 1 & Reviewer 2:

1. **According to the chemoproteomic analysis conducted to assess MYST inhibitor selectivity, KAT6A and KAT6B were not detected...In contrast, biochemical assays identified KAT6A and KAT6B as the most potent targets of PF-9363. This discrepancy highlights a significant limitation...While a brief explanation for this discrepancy is provided, further validation is necessary, particularly to investigate the low abundance of KAT6A/B.**

We thank the reviewer for this comment. While all chemoproteomic methods have sampling biases, we agree it is disappointing that our H3K14-CoA resin did not capture endogenous KAT6A or KAT6B in these experiments. To further validate that this is due to the low abundance of KAT6A and KAT6B in our samples, we performed additional analyses and experimentation.

- **Analysis:** First, we analyzed DepMap and compared the expression of KAT6A and KAT6B to other MYST family members. In the figure below, points lying above the diagonal represent cell lines in which the MYST family member defined on the y-axis (KAT5, KAT7, or KAT8) is expressed more strongly than KAT6A (top row) or KAT6B (bottom row). Of note, in almost every cell line of >1000 surveyed, KAT6A and KAT6B are more lowly expressed than other MYST family members.

Figure S2B: Analysis of the relative KAT6A (top) and KAT6B (bottom) expression compared to other MYST family members (KAT5, KAT7, KAT8) across the DepMap cell line panel.

This is consistent with our whole proteome proteomics analysis indicating KAT6A and KAT6B are of insufficient abundance to be detected in HeLa nuclear extracts. This analysis has been included as revised Figure S2.

- **Experiment:** To assess whether H3K14-CoA resin is capable of capturing KAT6A, we spiked full-length recombinant KAT6A into the nuclear extracts and compared proteins captured by an H3K14-CoA resin to background proteins captured by capped beads. This analysis verified that H3K14-CoA resin can significantly enrich KAT6A and that when KAT6A is present in these samples it is the most strongly competed by PF-9363 and its clinical analogue PF-8144. This data is described in the revised text and also included as Table S3.

“Comparing expression of MYST genes in DepMap indicated KAT6A/B are generally more weakly expressed than other family members (**Fig. S2B**) To address this ‘blind spot’ and assess the ability of chemoproteomics to assess selectivity across the MYST family, we performed a pilot in which we spiked recombinant KAT6A into nuclear extracts and measured competition by PF-9363 or PF-8144 at a single concentration (1 μ M). KAT6A was the most potently competed CoA-binding protein, followed by KAT7 (**Table S3; Fig. S3**).”

2. **Chemoproteomic analysis works on active site competition. How will the allosteric side competitors be addressed?**

This is an important point. All of our chemoproteomic analyses of MYST inhibitor use a CoA capture matrix; as such they are expected to only sample only interactions that are competitive with the CoA-binding active site (no CoA-binding allosteric sites of KATs are currently known). This has been specified in the text.

- “One caveat is that this approach exclusively reports on orthosteric, as opposed to allosteric, inhibition.”
3. **The quality and exposure of the H4 western blot presented in Figure 4 could be improved to enhance clarity and interpretability.**

As part of our revision experiments we re-ran the dose-dependent treatment of MCF-7 cells with PF-9363 and confirmed it does not affect overall levels of histone H4. The updated blot is included in the revised Figure 4.

4. **A better explanation is recommended on how FOXK2 is relevant for this paper and AN IP experiment to validate the *in Silico* data.**

We thank the reviewer for this suggestion. FOXK2 is primarily relevant to this study as a demonstration of how our chemoproteomic approach can be leveraged to identify new candidate members of MYST complexes. Specifically, FOXK2 emerged as one of only two proteins (out of >1000 captured) that were not previously annotated as MYST complex members yet displayed dose-dependent competition patterns highly similar to known KAT8 NSL complex proteins. This discovery validates two key methodological advances presented in our work: (1) that clustering of chemoproteomic competition data can systematically identify co-regulated protein assemblies, and (2) that AlphaFold-based screening can prioritize which of these candidate interactions are most likely to be genuine based on structural compatibility. While the specific biology that FOXK2 mediates within the NSL complex is beyond the scope of this paper, we have noted its documented roles in the literature. On a methodological level, the value of this strategy for MYST complex characterization is that it does not require genetic tags and can potentially be applied to native cells and tissues. This has been specified in the text.

- **Rationale:** “Applying MYST inhibitors as precision chemical probes requires not only defining their selectivity, but also the composition of the protein assemblies they engage”
- **Significance of method:** “This chemoproteomic approach is complementary to existing approaches for protein-protein interaction discovery but is distinct from many in that it does not require a genetic tag. This raises the possibility of using it to analyze the composition of KAT complexes and native cells and tissues”

Regarding validation, we agree on the importance of this experiment and have performed it as requested. Specifically, we ectopically-expressed FLAG-tagged FOXK2, which we found was able to co-immunoprecipitate both Myc-tagged OGT (the

known NSL complex member predicted by Alphafold to interact most strongly with it) and, notably, endogenous KAT8 itself (Figure 3E-F). To understand the specificity of this experiment we also explored co-immunoprecipitation of WDR5, another candidate interactor. In this case we saw higher background (enrichment in controls lacking FLAG-FOXK2), as well as an abrogation of the interaction under high salt wash conditions, indicative of non-specific or weak association of WDR5 and FOXK2.

Fig. 3 (E) Co-immunoprecipitation of FLAG-FOXK2 and Myc-OGT. Left: standard wash; Right: High salt wash. OGT interaction persists in presence of salt. (F) Co-immunoprecipitation of FLAG-FOXK2 and WDR5-Myc. Left: standard wash; Right: High salt wash. WDR5 interactions were salt-labile, suggestive of indirect or low-affinity contacts. In both cases FLAG-FOXK2 is found to co-immunoprecipitate endogenous KAT8.

These results confirm that FOXK2 physically associates with the KAT8 complex, as predicted from our chemoproteomic competition data, and that there is some specificity to the interaction. We note that this finding is consistent with previous reports showing NSL complex members (KANSL1/KANSL1L) co-immunoprecipitate with FOXK2, further supporting fractional occupancy of this transcription factor within the MYST complex. These data have been added to the revised manuscript.

- “Given the intrinsic uncertainty in predicting interactions of disordered regions, we experimentally evaluated the ability of OGT and WDR5 to interact with FOXK2 in co-immunoprecipitation assays (Fig. 3E-F). Ectopically-expressed FOXK2-FLAG captured using an anti-FLAG resin was able to co-enrich a Myc-tagged OGT using both basal and high salt wash conditions (Fig. 3E). An identical procedure was not able to co-precipitate WDR5-Myc using high salt wash conditions (Fig. 3F). STRING analysis revealed OGT has been previously linked to FOXK2 as a member of the polycomb repressive deubiquitinase (PR-DUB) complex.^{37, 38} Interestingly, FLAG-FOX2 by itself was able to co-immunoprecipitate endogenous KAT8. Previous studies have observed that members of the NSL complex (KANSL1/KANSL1L) co-immunoprecipitate with FOXK2,³⁹ further supporting the ability of this transcription factor to fractionally occupy the MYST complex in pulldown experiments.”
5. **To enable a direct comparison of the dose-dependent effects of the inhibitors on MYST-dependent biomarkers, MCF-7 cells were treated with varying concentrations. To validate the specificity of these effects, parallel treatment of MYST knockdown cells with the inhibitors is recommended.**

We appreciate this thoughtful suggestion. Because our chemoproteomic experiments did not sample endogenous KAT6A/B, the original manuscript did not display full concordance between the chemoproteomic target occupancy and cellular biomarker profiling data, which we agree would more intuitively link each MYST enzyme and its cognate histone mark. To address this, we have performed two additional experiments. **First**, we conducted KAT6A spike-in experiments demonstrating that PF-9363 competes KAT6A more potently than any other CoA-binding protein in chemoproteomic assays, establishing complete concordance between target occupancy and biomarker hierarchies (Table S3, Figure S3).

Figure S3. (A) Proteome-wide competition analysis of PF-9363 competition in HeLa nuclear extracts with exogenous KAT6A spiked in ($n = 3$ biological replicates). Nuclear extracts were pre-incubated at the specified concentration (2 h, 4 °C) prior to KAT affinity capture. MYST KATs and interactors are color-coded according to complex. (B) Identical proteome-wide competition analysis performed with PF-8144 (aka prifetrastat).

Second, we note that our biomarker assignments are fully consistent with genetic perturbation studies conducted by Sharma et al. (ref 9 of manuscript, Supplementary Figure 4 of Sharma et al.), who demonstrated that either double KO of KAT6A/B or single KO of BRPF1 (a scaffolding protein used by both KAT6A/B complexes) selectively impacts H3K23Ac but not H3K14Ac. Our dose-dependent biomarker data recapitulate this selectivity, demonstrating that at the lowest concentrations, dual

KAT6A/B occupancy downregulates H3K23Ac, followed by sequential inhibition of H3K14Ac (KAT7), H4K16Ac (KAT8), and H2A.Zac (KAT5) at progressively higher concentrations. Additional cellular profiling data (added in this revision) further expands this biomarker profile to the clinical candidate PF-8144.

The combination of (1) KAT6A spike-in experiments confirming chemoproteomic competition, (2) full concordance between target occupancy and biomarker inhibition hierarchies, and (3) consistency with published genetic knockdown studies collectively provide robust validation of biomarker specificity. While knockdown experiments could provide additional confirmation of previous results from the known literature, the agreement of the chemoproteomic and biomarker data combined with references to previous studies helpfully consolidates the data in the field and establishes the specific KAT enzyme responsible for each biomarker with high confidence. We have clarified this logic when laying out the goals of the study in our revised manuscript.

- “To help researchers accurately deploy MYST inhibitors as chemical probes and clinical agents, there remains an unmet need to: (1) define their proteome-wide target engagement profiles, (2) consolidate our understanding of what histone acetylations can be used as biomarkers of target engagement in cells and tissues, (3) analyze the comparative specificity and potency of MYST inhibitors across as broad a spectrum of chemotypes as possible, and (4) characterize the phenotypic relevance and any polypharmacology they may display.”
6. **A clearer explanation is recommended to clarify the relevance of FOXK2 in the context of this study. Additionally, an immunoprecipitation (IP) experiment would be valuable to experimentally validate the in silico findings.**

This comment is addressed in response #4 (above).

7. **Reframe the sentence: Target engagement was assessed using the panel of MYST-dependent histone acetylation biomarkers defined above. Analyzing the first-generation sulfonohydrazide WM-8014, we observed it was old less potent than PF-9363 but afforded similarly complete inhibition of KAT6A/B – as assessed by inhibition of H3K23Ac – at 1 μM (Fig. 5B)**

We have revised this sentence to clarify it as suggested.

- **Shifting our attention to the first generation sulfonohydrazide WM-8014, we observed it was old less potent than PF-9363 but afforded similarly complete inhibition of KAT6A/B – as assessed by inhibition of H3K23Ac – at 1 μM (Fig. 5B).**
8. **Typing error - Our chemoproteomic assay cannot distinguish whether this reflects preferential capture of the BRPF complex by the affinity resin, preferential engagement of the ~~the~~ JADE complex by PF-9363, or association of BRPF3 with another MYST complex.**

This has been corrected.

9. **The dose-dependent inhibition (e.g., 10–30 μM) of KAT7/KAT8/KAT5 may not be pharmacologically relevant.**

We thank the reviewer for this important comment, which has prompted us to better articulate the context and rationale for characterizing high-dose MYST inhibitor effects. First, point out that there is substantial evidence indicating that dose-dependent inhibition of KAT7 and KAT8 can be pharmacologically relevant in multiple contexts.

KAT7 engagement is therapeutically relevant: MacPherson et al. (ref. 14) used WM-3835 - which we now show inhibits KAT6A/B at low concentrations and KAT7 at higher concentrations - to probe KAT7-dependent oncogenic transcription in acute myeloid leukemia. Importantly, the phenotypes they observed required dual KAT6/KAT7 inhibition to phenocopy aspects of gene knockout, demonstrating that the ability of MYST inhibitors to hierarchically engage multiple targets can act as a relevant driver of their physiological effects. During preparation of this manuscript, Perner et al. (ref. 53) reported that dual KAT6/KAT7 inhibition using PF-9363 synergizes with Menin inhibitors to overcome primary and acquired drug resistance in MLL-rearranged leukemia. This timely work illustrates how an understanding of dose-dependent target occupancy can be used to design strategies that rationally leverage MYST inhibitor polypharmacology for therapeutic benefit. These findings are now described explicitly in the revised text:

- “One implication of the data above is that the ability of MYST inhibitors to hierarchically engage multiple targets can act as a relevant driver of their physiological effects. Specifically, even though KAT6A/B is the primary target of WM-3835, its ability to secondarily engage KAT7 allows it to phenocopy aspects of gene knockout.¹ Put another way, while WM-3835 is not a specific chemical probe of KAT7, it can usefully modulate KAT7-dependent phenotypes.”
- “During preparation of this manuscript, a study by Perner et al. reported that dual KAT6/KAT7 inhibition using PF-9363 synergizes with Menin inhibitors to overcome primary and acquired drug resistance in MLL-rearranged leukemia.² This timely work illustrates how an understanding of dose-dependent target occupancy can be used to design strategies that rationally leverage MYST inhibitor polypharmacology for therapeutic benefit.”

PF-9363 can be applied as a KAT8 modulator. The reviewer’s comment also made us aware that we had failed to adequately frame the utility of the KAT8 experiments. While WM-3835 is not a chemical probe of KAT7 (due to its coincident inhibition of KAT6A/B) it can be used to modulate KAT7 activity when applied at sufficient concentrations. A key novelty of our study – not found elsewhere in the literature - is that PF-9363’s potency allows it to be similarly applied to modulate KAT8. This knowledge is useful to researchers interested in modulating KAT8-mediated H4K16Ac with a drug-like small molecule. To illustrate this we performed a simple study, showing a KAT8-mediated phenotype (acute antiproliferative activity) correlates with loss of the biomarker (H4K16Ac) only at concentrations of PF-9363 (>5-10 μ M) sufficient to occupy the KAT8 active site. This has been clarified in the text.

- “Considering the biomarker profile of PF-9363 (**Fig. 5B**), we wondered if this strategy could be similarly harnessed to alter a KAT8-dependent phenotype. As a proof-of-concept we focused on acute growth inhibition. Our rationale was that, when used to inhibit KAT6A/B and/or KAT7, MYST inhibitors require prolonged administration (~8-21 days) to suppress senescence⁹ or growth.^{9, 45} Unlike KAT6A/B and KAT7, KAT8 is a universally essential enzyme. Thus, upon treating cells with concentrations of PF-9363 sufficient to engage KAT8, we would expect acute growth defects that are relatively independent of cell lineage compared to other KAT inhibitors.”

Answering an additional question implied by the reviewer –whether inhibition of KAT7 or KAT8 is relevant for PF-9363 or PF-8144 in clinical settings—will require analysis of patient samples which we were unable to access in the time frame of this revision. Regardless, it is important to recognize this possibility exists due to the high potencies of these molecules and similar active site architectures of MYST family members, a feature our work comprehensively highlights for the first time.

10. The author can use consistent naming for KATs throughout (either KAT6A/B or KAT6A/KAT6B). (Introduction, Line 24)

Great comment, we agree! This has been addressed.

Reviewer #3:

1. **... the narrative and motivation for the study are unclear....The paper would benefit from being written to be more accessibly so that non-experts can understand the work better...For example, if the authors want the wider scientific community to heed their warnings about the commercial inhibitors specificities.**

We are grateful to the reviewer for this blunt albeit highly helpful assessment. Reviewing the initial manuscript, it became apparent that our introduction had implied rather than outright stated the motivation for our studies of PF-9363. This also made us aware that we could clarify what we hoped to learn in our studies of FOXK2 and KAT8 inhibition. This prompted significant revision to the text.

- **Introduction – motivation for overall study:** To help researchers accurately deploy MYST inhibitors as chemical probes and clinical agents, there remains an unmet need to: (1) define their proteome-wide target engagement profiles, (2) consolidate our understanding of what histone acetylations can be used as biomarkers of target engagement in cells and tissues, (3) analyze the comparative specificity and potency of MYST inhibitors across as broad a spectrum of chemotypes as possible, and (4) characterize the phenotypic relevance and any polypharmacology they may display.”

- **Results – motivation for FOXK2 study:** Applying MYST inhibitors as precision chemical probes requires not only defining their selectivity, but also the composition of the protein assemblies they engage
- **Results – motivation for KAT8 study:** “One implication of the data above is that the ability of MYST inhibitors to hierarchically engage multiple targets can act as a relevant driver of their physiological effects. Specifically, even though KAT6A/B is the primary target of WM-3835, its ability to secondarily engage KAT7 allows it to phenocopy aspects of gene knockout.¹⁴ Put another way, while WM-3835 is not a specific chemical probe of KAT7, it can usefully modulate KAT7-dependent phenotypes. Considering the biomarker profile of PF-9363 (Fig. 5B), we wondered if this strategy could be similarly harnessed to alter a KAT8-dependent phenotype. As a proof-of-concept we focused on acute growth inhibition. Our rationale was that, when used to inhibit KAT6A/B and/or KAT7, MYST inhibitors require prolonged administration (~8-21 days) to suppress senescence⁸ or growth.^{9, 45} Unlike KAT6A/B and KAT7, KAT8 is a universally essential enzyme. Thus, upon treating cells with concentrations of PF-9363 sufficient to engage KAT8, we would expect acute growth defects that are relatively independent of cell lineage compared to other KAT inhibitors.”

We’ve also attempted to re-iterate the motivation and clarify the take-homes in the discussion.

- **Discussion – motivation for overall study:** “Targeting lysine acetylation via inhibition of KAT activity is an emerging paradigm in oncology,⁴⁴ with several compounds now in clinical evaluation. Critical to identifying new therapeutic contexts for these agents is the proper interpretation of their preclinical effects. Towards that end, here we report a comparative analysis of drug-like MYST acetyltransferase inhibitors.”
 - **Discussion – take-home for MYST probes v. modulators:** “These findings highlight the distinction between applying these compounds as chemical probes – in which they target a single enzyme e.g. KAT6A/B – versus modulators, in which their pharmacology is leveraged to engage additional targets (e.g. KAT7/KAT8) at higher concentrations.”
 - **Discussion – take-home for MYST inhibitor usage:** “As a general rule we recommend monitoring these histone marks when deploying MYST inhibitors in novel systems and/or at escalated doses. This straightforward experiment, which can be carried out using commercial antibodies, provides a powerful measure to ensure that mechanistic effects are properly attributed to a given KAT enzyme’s inhibition.”
2. **The authors report that the chemoproteomics approach did not identify KAT6 but that this is because KAT6 is low/not expressed in HeLa cells. If KAT6 were expressed, would this change the IC50 concentrations at which PF-9363 might inhibit other KATs?**

The reviewers raise an interesting point. It is possible that if there were extremely high concentrations of KAT6A/B present in nuclear extracts during our chemoproteomic experiments, it may deplete PF-9363 to such an extent that higher concentrations were necessary to inhibit capture of KAT7 by our H3K14-CoA affinity resin, widening the specificity window. In reality, this is not a large issue, as we do not use chemoproteomics to determine quantitative IC₅₀ values. Our results from the chemoproteomic experiment using spike-in protein indicates that even in the presence of supraphysiological concentrations of KAT6A ‘host,’ KAT7 is competed (Figure S3). With this additional result, we now observe full concordance between chemoproteomic target occupancy and cellular biomarker profiling data, intuitively linking each MYST enzyme to its cognate histone mark.

3. **The authors claim that their approach demonstrates a strategy for identifying new candidate interactions partners of multiprotein KAT complexes. Some additional discussion about why their approach might be better than other existing strategies people use to unbiasedly investigate novel protein-protein interactions in multiprotein complexes would be helpful. It seems like...they might need to actual validate some aspect of their...model of the interactions?**

Here we apologize if our discussion was difficult to interpret. We view KAT chemoproteomic capture/PF-9363 competition as a complementary rather than “better” strategy for MYST protein-protein interaction discovery. While it lacks the specificity of traditional approaches (e.g. tagging specific MYST complex members with affinity tags such as Myc or Alfa) it holds the advantage of 1) probing multiple complexes simultaneously, and 2) not requiring cell line generation. This has been better specified in the Discussion.

- **Significance of PPI discovery strategy:** “This chemoproteomic approach is complementary to existing approaches for protein-protein interaction discovery but is distinct from many in that it does not require a genetic tag. This raises the possibility of using it to analyze the composition of KAT complexes and native cells and tissues”

Regarding validation, as specified in the response to Reviewer #1 and #2, we agree on the importance of this experiment and have performed it as requested. To re-iterate, we ectopically-expressed FLAG-tagged FOXK2, which we found was able to co-immunoprecipitate both Myc-tagged OGT (the known NSL complex member predicted by AlphaFold to interact most strongly with it) and, notably, endogenous KAT8 itself (Figure 3E-F). To understand the specificity of this experiment we also explored co-immunoprecipitation of WDR5, another other candidate interactor. In this case we saw higher background (enrichment in

controls lacking FLAG-FOXK2), as well as an abrogation of the interaction under high salt wash conditions, indicative of non-specific or weak association of WDR5 and FOXK2.

Fig. 3 (E) Co-immunoprecipitation of FLAG-FOXK2 and Myc-OGT. Left: standard wash; Right: High salt wash. OGT interaction persists in presence of salt. (F) Co-immunoprecipitation of FLAG-FOXK2 and WDR5-Myc. Left: standard wash; Right: High salt wash. WDR5 interactions were salt-labile, suggestive of indirect or low-affinity contacts. In both cases FLAG-FOXK2 is found to co-immunoprecipitate endogenous KAT8.

These results confirm that FOXK2 physically associates with the KAT8 complex, as predicted from our chemoproteomic competition data, and that there is some specificity to the interaction. We note that this finding is consistent with previous reports showing NSL complex members (KANSL1/KANSL1L) co-immunoprecipitate with FOXK2, further supporting fractional occupancy of this transcription factor within the MYST complex. These data have been added to the revised manuscript.

- “Given the intrinsic uncertainty in predicting interactions of disordered regions, we experimentally evaluated the ability of OGT and WDR5 to interact with FOXK2 in co-immunoprecipitation assays (Fig. 3E-F). Ectopically-expressed FOXK2-FLAG captured using an anti-FLAG resin was able to co-enrich a Myc-tagged OGT using both basal and high salt wash conditions (Fig. 3E). An identical procedure was not able to co-precipitate WDR5-Myc using high salt wash conditions (Fig. 3F). STRING analysis revealed OGT has been previously linked to FOXK2 as a member of the polycomb repressive deubiquitinase (PR-DUB) complex.^{37, 38} Interestingly, FLAG-FOXK2 by itself was able to co-immunoprecipitate endogenous KAT8. Previous studies have observed that members of the NSL complex (KANSL1/KANSL1L) co-immunoprecipitate with FOXK2,³⁹ further supporting the ability of this transcription factor to fractionally occupy the MYST complex in pulldown experiments.”

The authors perform the MCF-7 experiments in Figure 4 to establish the physiological relevance of MYST complex inhibition. Could the authors comment on the justification or physiological relevance of the PF-9363 concentrations that they used? Is this effectively a similar experiment to Figure 2 except cells were treated directly so it's a functional readout of Kac peptides rather than protein complexes?

We appreciate this question, as it re-emphasizes the logic connecting our chemoproteomic and cell experiments. Addressing the second question first: yes, the reviewer is correct that Figure 4 represents the cellular counterpart to Figure 2 - whereas chemoproteomics measures target occupancy in nuclear extracts, the histone biomarker analysis measures the functional consequences of inhibitor treatment in living cells. The goal was to demonstrate concordance between biochemical target engagement and cellular activity across the same concentration range.

Regarding the justification for the PF-9363 concentrations tested (0.1-30 μ M), examining effects across this dose range allowed our study to address two related objectives: **1) Determining concordance between chemoproteomic and cellular experiments.** Prior to our chemoproteomic studies, researchers had not described the ability of PF-9363 to engage KAT8- or KAT5-containing complexes in cells or proteomes. Conceivably, this phenomenon could be limited to chemoproteomic measurements. Examining the full dose range allowed us to demonstrate PF-9363 can also engage these targets in living cells. **2) Establishing a strategy for pharmacologically modulating KAT8.** Unlike KAT6A/B and KAT7, KAT8 is a universally essential enzyme with limited available chemical tools. We demonstrate that PF-9363 (and PF-8144) uniquely among the compounds tested possess the ability to engage KAT8 at elevated concentrations (10-30 μ M), providing a strategy for using these compounds to modulate KAT8 activity and probe KAT8-dependent phenotypes.

Regarding the physiological relevance of these concentrations, we do not yet know what exposures are achieved in patient tissues during clinical dosing of PF-8144. The Phase 1 clinical trial (ref. 10 of text) demonstrated reduced H3K23Ac (KAT6A/B biomarker) in patient samples, but whether KAT7 or KAT8 biomarkers were affected was not determined. A key point of our manuscript is to provide the framework and biomarker panel necessary to address this question - researchers and clinicians should monitor H3K14Ac (KAT7) and H4K16Ac (KAT8) alongside H3K23Ac (KAT6A/B) to calibrate dosing and fully understand on-target and potential off-target engagement. This comprehensive profiling data will have utility in enabling the field to properly interpret mechanism of action and rational dosing strategies.

- 4. I understand for the NCI-60 cells in Figure 6 why one might want to broaden their inhibitor studies to validate them in more cells, but the reasoning that the long-term growth changes are known but not the short-term effects feels a little underdeveloped. Can the authors expand on why the short-term effects are important too, what new insights these experiments provide, and where we go from here?**

We thank the reviewer for this comment, which (as described above) made us realize we failed to adequately frame the rationale for these experiments. The key insight is that PF-9363 can be applied as a KAT8 modulator - not a selective probe, but a tool to pharmacologically perturb KAT8 activity and probe KAT8-dependent phenotypes. Since KAT8 is universally essential, we hypothesized its inhibition would cause acute growth defects, whereas KAT6/KAT7 inhibition requires prolonged treatment (8-21 days) to affect growth or senescence. Our short-term (48 hour) NCI-60 data support this: concentrations affecting primarily KAT6/KAT7 (<1 μ M) are well-tolerated, while growth inhibition occurs at 10-100 μ M where KAT8 is also engaged. In BT-549 cells, acute toxicity (GI50 \sim 7 μ M) coincides with inhibition of KAT6A/B (H3K23Ac), KAT7 (H3K14Ac), and KAT8 (H4K16Ac).

While toxicity admittedly represents a 'blunt' phenotype, the significance of this finding is that it demonstrates a strategy where researchers can contrast WM-1119 (KAT6), WM-3835 (KAT6/KAT7), and PF-9363 (KAT6/KAT7/KAT8) to dissect individual enzyme contributions. Additionally, since H4K16Ac affords the capability (unique among histone acetylations studied to date) to modulate nucleosome structure directly, the ability to pharmacologically perturb this mark has utility beyond growth inhibition. Future work could use PF-9353 to probe H4K16Ac-dependent on chromatin structure at short time points, or test Helin and coworkers' hypothesis that KAT8-low tumors are uniquely vulnerable to KAT8 inhibition, leveraging the differential pharmacology we've defined. Establishing these compounds as experimental KAT8 modulators (and the ability to use WM-3835 as a non-KAT8 perturbing comparison compound) represents a valuable addition to the KAT chemical biology toolkit.

Reviewer #4:

- 1. Line 24 of the abstract contains a typo: "histone acetylation biomarkers" should be corrected.**

This has been corrected

- 2. Figure 2D: If feasible, inclusion of a KAT5 immunoblot would help corroborate the chemoproteomics results.**

We have opted not to include this data based on resource/personnel considerations as its conclusions are already corroborated by both the chemoproteomic and biomarkers assays.

- 3. Pulldown assay limitations: The authors acknowledge the failure to recover KAT6A/B but do not experimentally address this limitation. Could alternative fractionation protocols or overexpression of KAT6A/B be tested?**

Rather than overexpress KAT6A/B we chose to overcome this limitation using a spike-in approach. As described above, full-length recombinant KAT6A was spiked into HeLa nuclear extracts and exposed to our H3K14-CoA capture resin in the presence or absence of PF-9363 or PF-8144 (1 μ M). Proteomic analyses confirmed 1) that our H3K14-CoA resin is capable of capturing KAT6A and 2) that both PF-9363 and PF-8144 compete KAT6A more potently than any other CoA-binding protein, followed by KAT7 and its complex members (Table S3; Figure S3). This experiment creates concordance between our chemoproteomic results and the biomarker-supported selectivity hierarchy.

- 4. Figure 3E and 3F are labeled "AF2 multimer prediction...", while the legend refers to "AF3-Multimer prediction...". This inconsistency should be corrected.**

This has been corrected. In order to prioritize experimental data, the predicted structures now reside in Figure S4.

- 5. FOXK2 association: Dose-dependent depletion of FOXK2 upon PF-9363 treatment implies a potential association with MYST complexes, possibly through WDR5 or OGT. The authors should consider confirming this interaction via co-immunoprecipitation**

We thank the reviewer for this comment. We agree on the importance of this experiment which was commonly requested by all four reviewers. To re-iterate: we ectopically-expressed FLAG-tagged FOXK2, which we found was able to co-immunoprecipitate both Myc-tagged OGT (the known NSL complex member predicted by AlphaFold to interact most strongly with it) and, notably, endogenous KAT8 itself (Figure 3E-F). To understand the specificity of this experiment we also explored co-immunoprecipitation of WDR5, another other candidate interactor. In this case we saw higher background (enrichment in controls lacking FLAG-FOXK2), as well as an abrogation of the interaction under high salt wash conditions, indicative of non-specific or weak association of WDR5 and FOXK2.

Fig. 3 (E) Co-immunoprecipitation of FLAG-FOXP2 and Myc-OGT. Left: standard wash; Right: High salt wash. OGT interaction persists in presence of salt. (F) Co-immunoprecipitation of FLAG-FOXP2 and WDR5-Myc. Left: standard wash; Right: High salt wash. WDR5 interactions were salt-labile, suggestive of indirect or low-affinity contacts. In both cases FLAG-FOXP2 is found to co-immunoprecipitate endogenous KAT8.

These results confirm that FOXP2 physically associates with the KAT8 complex, as predicted from our chemoproteomic competition data, and that there is some specificity to the interaction. We note that this finding is consistent with previous reports showing NSL complex members (KANSL1/KANSL1L) co-immunoprecipitate with FOXP2, further supporting fractional occupancy of this transcription factor within the MYST complex. These data have been added to the revised manuscript.

- “Given the intrinsic uncertainty in predicting interactions of disordered regions, we experimentally evaluated the ability of OGT and WDR5 to interact with FOXP2 in co-immunoprecipitation assays (Fig. 3E-F). Ectopically-expressed FOXP2-FLAG captured using an anti-FLAG resin was able to co-enrich a Myc-tagged OGT using both basal and high salt wash conditions (Fig. 3E). An identical procedure was not able to co-precipitate WDR5-Myc using high salt wash conditions (Fig. 3F). STRING analysis revealed OGT has been previously linked to FOXP2 as a member of the polycomb repressive deubiquitinase (PR-DUB) complex.^{37, 38} Interestingly, FLAG-FOX2 by itself was able to co-immunoprecipitate endogenous KAT8. Previous studies have observed that members of the NSL complex (KANSL1/KANSL1L) co-immunoprecipitate with FOXP2,³⁹ further supporting the ability of this transcription factor to fractionally occupy the MYST complex in pulldown experiments.”

6. **Line 40 of the section “Assessing target engagement by orthogonal MYST inhibitor chemotypes” uses the phrase “old less potent...”—this wording is awkward and should be revised for clarity.**

This has been revised.

Summary of experiments performed for revision

- Re-performed inhibitor treatments and KAc biomarker analysis with PF-9363 in MCF-7 cells (third biological replicate)
- Added inhibitor treatments and KAc biomarker analysis for clinical candidate PF-8144
- Added inhibitor treatments and KAc biomarker analysis for additional MYST inhibitor chemotypes
- Co-immunoprecipitation of FOXP2 with OGT
- Co-immunoprecipitation of FOXP2 with WDR5
- Co-immunoprecipitation of FOXP2 with endogenous KAT8
- LC-MS proteomic analysis of chemoproteomic competition using recombinant KAT6A spike-ins
- Chemoproteomic analysis of clinical candidate PF-8144
- DepMap expression analysis comparing KAT6A/B abundance to other MYST family members
- Added discussion of previous studies (BPTF knockdown) that independently validate biomarker assignments

Outcome/value of revision experiments

- Improved concordance between chemoproteomic target occupancy and cellular target engagement biomarkers
- Added discussion of previous studies (BPTF knockdown) that independently validate biomarker assignments
- Experimental confirmation of FOXP2 association with KAT8 complex members (OGT, endogenous KAT8)
- Validation that chemoproteomic approach is applicable to KAT6A/B
- Re-confirmation of PF-9363 dose-response data with third biological replicate
- Extension of hierarchical inhibition framework to clinical candidate PF-8144 (Figure 5)
- Strengthened rationale for KAT8 studies and distinction between chemical probes versus modulators

Our revision has substantially expanded the molecular characterization of MYST inhibitor pharmacology through multiple new experiments totaling significant resources and effort. These data have reinforced and extended, rather than altered, our fundamental conclusions regarding the hierarchical target engagement of MYST family members by drug-like inhibitors. The revision has particularly strengthened: (1) the concordance between chemoproteomic and cellular biomarker data through

KAT6A spike-in experiments, (2) the validation of the FOXK2-NSL interaction through co-immunoprecipitation, and (3) the conceptual framework distinguishing selective chemical probes from multi-target modulators.

Given the significant investment already made in these revisions (4 months of personnel time) and the uncertainty regarding the extent to which ongoing disruptions to federal research operations will permit us to commit additional resources to experimental work beyond this revision, we respectfully ask the reviewers to consider both these extensive new data and their initial positive assessment of our work's importance and therapeutic implications when making their recommendation.

Once again, we are grateful to the referees for their helpful comments and to you for all your efforts on our behalf.

Sincerely,

Jordan L. Meier
Senior Investigator
Chemical Biology Laboratory,
National Cancer Institute, NIH